# The Untapped Potential of Hairy Root Cultures and Their Multiple Applications

**DOI:** 10.3390/ijms252312682

**Published:** 2024-11-26

**Authors:** Iman Mirmazloum, Aleksandar K. Slavov, Andrey S. Marchev

**Affiliations:** 1Department of Plant Physiology and Plant Ecology, Institute of Agronomy, Hungarian University of Agriculture and Life Sciences, Ménesi Str. 44, 1118 Budapest, Hungary; mirmazloum.seyediman@uni-mate.hu; 2Department of Ecological Engineering, University of Food Technologies Plovdiv, 26 Maritsa Blvd., 4002 Plovdiv, Bulgaria; alex_slavov@uft-plovdiv.bg; 3Laboratory of Eukaryotic Cell Biology, Department of Biotechnology, The Stephan Angeloff Institute of Microbiology, Bulgarian Academy of Sciences, 139 Ruski Blvd., 4000 Plovdiv, Bulgaria

**Keywords:** specialized metabolites, hairy roots, *Rhizobium rhizogenes*, elicitation, metabolic engineering, CRISPR/CAS9, bioprocessing, bioreactors, patents, industrial scale

## Abstract

Plants are rich sources of specialized metabolites, such as alkaloids, terpenes, phenolic acids, flavonoids, coumarins, and volatile oils, which provide various health benefits including anticancer, anti-inflammatory, antiaging, skin-altering, and anti-diabetic properties. However, challenges such as low and inconsistent yields, environment and geographic factors, and species-specific production of some specialized metabolites limit the supply of raw plant material for the food, cosmetic, or pharmaceutical industries. Therefore, biotechnological approaches using plant in vitro systems offer an appealing alternative for the production of biologically active metabolites. Among these, hairy root cultures induced by *Rhizobium rhizogenes* have firmed up their position as “green cell factories” due to their genotypic and biosynthetic stability. Hairy roots are valuable platforms for producing high-value phytomolecules at a low cost, are amenable to pathway engineering, and can be scaled up in bioreactors, making them attractive for commercialization. This review explores the potential of hairy roots for specialized metabolites biosynthesis focusing on biotechnology tools to enhance their production. Aspects of morphological peculiarities of hairy roots, the diversity of bioreactors design, and process intensification technologies for maximizing biosynthetic capacity, as well as examples of patented plant-derived (green-labeled) products produced through hairy root cultivation at lab and industrial scales, are addressed and discussed.

## 1. Introduction

Plants might be regarded as an almost unlimited resource and a biochemical factory for the biosynthesis of both primary (e.g., carbohydrates, amino acids, and lipids) and highly valuable secondary/specialized metabolites (SMs), including alkaloids, terpenes, phenolic acids, flavonoids, tannins, coumarins, quinones, and volatile oils, etc. [1,2]. The SMs comprise a vast range of low-molecular-weight compounds that are not essential in the survival of plants but provide defense against biotic factors (e.g., herbivores, fungi, and bacteria) and abiotic stresses (e.g., UV radiation, temperature, and drought), and are also attracting agents during pollination and seed dispersal [3]. These SMs have diverse beneficial biological activities important for humans, including anticancer, anti-inflammatory, antiaging, skin-altering, and anti-diabetic effects, among others, and are intensively incorporated in plant-derived cosmetic products, pharmaceuticals, nutraceuticals, or food additives [4]. Some of the widely used plant-derived active ingredients with enormous commercial value that are incorporated in pharmacologically important formulations or solely administered are paclitaxel, atropine, morphine, dopamine, and artemisinin [5]. Plant SMs possess several particular features, including complex and specifically arranged aromatic rings, chiral centers, and chemical entities. These characteristics make them not only valuable targets for drug discovery but also starting points for new synthetic drugs, though this process is often very complicated and economically challenging [6]. Hence, a quarter of the drugs approved by the Food and Drug Administration (FDA) and/or the European Medicinal Agency (EMA) are plant-originated [7,8]. However, obstacles such as low production yield of the SMs, slow plant growth, dependence on anthropological, environmental, ecological, and demographic factors, as well as difficulties in the purification of the SMs and the potential presence of toxic substances, result in unavoidable fluctuations in yield and quality [9]. The broad diversity of plants and plant-derived natural products has created a considerable imbalance between their demand and availability. Consequently, there is growing global interest in establishing more effective alternative sources of important phytomolecules to address these challenges [10,11].

In the past decades, there has been a growing interest in plant in vitro systems as an appealing alternative for the production of biologically active metabolites. Plant in vitro systems offer opportunities for the production of high-value and marketable molecules through diverse biotechnological methods, possessing the following advantages: (1) production of SMs independent of internal or external factors; (2) consistency in yield and quality of natural products between different batches; (3) production under aseptic controlled conditions, improving the biosafety of the final product (free from environmental or genetic contamination); and (4) reduced production cycle time compared to the whole plant [12]. Therefore, in vitro plant-based systems are progressively considered for the large-scale production of valuable medicinal or cosmetic ingredients at lower cost [13]. The most frequently used plant in vitro systems for SMs biosynthesis have been the hairy roots (HRs) and plant cell suspension cultures. The induction and development of cell suspensions requires exogenous supplementation of plant growth regulators, and due to its more simplified bioreactor design, they are still preferred for the production of SMs at lab- or large scales. However, the somaclonal variation and genesis of undifferentiated tissues often results in limited growth and inconsistent yields of natural products [14]. Nevertheless, the production of SMs by HRs has garnered much more attention, owing to their extraordinary advantages, such as inherent genotypic, phenotypic, and biosynthetic stability [15].

Recently published reviews have limited focus on HRs mainly based on the possibilities to increase the SMs biosynthesis capacity, presenting familiar biosynthetic pathways of phenolic acids (salvianolic and rosmarinic acids) or terpenoid indole alkaloids [16,17]. Other review articles present in detail the processes of HR induction and applications [18] or have been focused on definite plant species, indicating their perspectives for biosynthesis of SMs and scale-up cultivation possibilities [19,20,21].

This review attempts to unravel the untapped potential of HRs for biosynthesis of high-value SMs and bring new insights in the biotechnology toolkit utilized to enhance SM production. An overview of various strategies to improve SM production is presented, including nutrient medium optimization, precursor feeding, biotic and abiotic elicitation, and molecular approaches such as metabolic engineering through the manipulation of single and/or multiple genes involved in the natural products’ biosynthesis pathways by CRISPR/Cas9 systems or the overexpression/suppression of transcription factors (TFs).

The review also covers some aspects of the HRs morphological peculiarities, the diversity of bioreactors design, and process intensification technologies for achieving maximum biosynthetic capacity. Special attention is given to the biosynthesis of green-labeled (plant-derived) products through HRs cultivation at both lab and industrial scales. A summary of the multiple applications and recent patents for production of SMs based on HRs cultivation is also highlighted. The workflow of this review is illustrated in Figure 1.

## 2. *Rhizobium rhizogenes* Mediated Genetic Transformation and Hairy Roots Untapped Potential

HRs are known to be induced by genetic-mediated transformation with the soil Gram-negative bacterium *Agrobacterium rhizogenes*, currently known as *Rhizobium rhizogenes*. Although the molecular events behind this process are not fully understood, it can generally be summed up in several stages: (1) chemotaxis or activation of the attachment, during which *R. rhizogenes* is attracted to plant tissues after wounding and triggers the secretion of phenolic compounds (acetosyringone) that are recognized by the bacteria; (2) processing of the T-DNA (transfer DNA) component of the root-inducing plasmid (Ri-plasmid) facilitated by virulence (vir) genes on the Ri-plasmid and the chromosomal virulence (chv) genes of bacterial chromosomal DNA; (3) incorporation of the T-DNA into the host plant genome; (4) expression of the T-DNA into the host plant genome; and (5) induction of the HRs from the wound site.

The T-DNA fragment contains a set of genes responsible for the synthesis of phytohormones, such as auxin and cytokinins, which drives the characteristic development of HRs as well as genes encoding for opine synthesis (products of condensation of amino acids with ketoacids or sugars used by *R. rhizogenes* as a source of carbon and nitrogen) [16,17]. Among the 18 open reading frames (ORFs) in the T-DNA region, the ‘*rol*’ (rooting locus) genes, *rolA*, *rolB*, *rolC*, and *rolD*, are essential for HRs formation. These oncogenes modulate plant cellular processes, including the regulation of phytohormone homeostasis, as well as auxin and cytokinin metabolism and signaling pathways [18,19]. Among these, *rolB* is particularly important for HRs induction. For example, *rolB* hydrolyses bound auxins resulting in increased intracellular level of indole-3-acetic acid, which is crucial for HRs induction [20]. In a loss-of-function study, knocking out the *rolB* genes rendered the Ri-plasmid avirulence [21]. Conversely, the *rolC* gene hydrolyses cytokinin conjugates, thereby liberating cytokinin that cumulatively enhance HRs growth when expressed with *rolA* and *rolB* [22]. *Rhizobium rhizogenes*-mediated HRs induction can be conducted using both in vitro and in planta inoculation techniques. The current methods for in vitro HRs induction could be classified as follows: (1) Direct plant inoculation, in which *R. rhizogenes* liquid culture is directly injected into the plant explant and (2) Co-cultivation method, which involves simultaneous cultivating of the plant explant with the bacterial suspension. To increase the transformation efficacy, treatments such as vacuum infiltration and sonication have also been used in the in vitro inoculation method. These techniques, as well as the main steps in HRs induction, were described in more detail elsewhere [16,18,22,23,24]. The in vitro techniques are performed in an aseptic, contamination-free environment, where the *R. rhizogenes* strain, type, and age of explants are influential [18,25]. On the opposite, in planta techniques do not require aseptic conditions, and therefore they are faster, less labor-intensive, and more cost-effective. The main techniques here are vacuum infiltration, *R. rhizogenes* injection, pollen tube-mediated gene transfer, floral dip, and floral spray methods described in more detail in [24,25] and are mostly based on the protocol of [26]. The *R. rhizogenes*-induced HRs in vitro have the potential to facilitate the generation of transgenic plants through spontaneous organ regeneration, organogenesis, or somatic embryogenesis. Finally, these transgenic plants can be propagated in vitro or transferred to the field [18,27]. However, when the aim of HRs induction is SMs biosynthesis, they are typically cultivated in shake-flasks, and further, the cultivation is scaled up into bioreactors [28], as illustrated in Figure 1. 

In conclusion, HRs typically exhibit rapid growth and lateral branching, abundant root hairs with no demand for plant hormones in the medium [16,17,23,24]. They are a suitable platform for the production of pharmaceutically valuable and complex SMs, recombinant proteins, as well as novel drugs associated with a wide range of medicinal plants, firming up their position as “green cell factories” [19]. Recombinant proteins produced by HRs are frequently secreted extracellularly, facilitating convenient purification in a well-defined, protein-deficient medium [25,26]. Amongst their many practical applications, HRs play an important role in metabolic engineering, bioreactor designing, biotransformation-mediated derivatization studies, phytoremediation, and plant-microbe interaction analyses, among others [27,28]. These versatile properties of HRs have progressively revealed numerous avenues of both fundamental and practical applications, ultimately enabling this technology to reach industrial platforms for the large-scale production of high-value-added molecules [29].

## 3. Plant Biotechnology Toolkit to Enhance Specialized Metabolite Production

Successful induction of HRs is influenced by several factors, including the type and development stage of the plant explant, the bacterial strain used, and the composition of culture medium [30]. The HR cultures are ideal systems for the stable and continuous production of SMs with relatively high yields. However, to meet commercial market demands, significant efforts have been invested in the selection of root cultures with higher production capacity, media composition, precursor feeding, elicitation, and metabolic engineering, as well as the development of scale-up strategies [31]. As a result, numerous HR cultures have been established for the biosynthesis of important phytochemicals, such as artemisinin from *Artemisia annua* L., camptothecin from *Camptotheca acuminata* Decne, resveratrol from *Arachis hypogaea* L., and indole alkaloids from *Catharanthus roseus* (L.) G.Don [32]. Additionally, HR cultures have been employed for the production of polyphenols, such as phenolic acids, flavonoids, and anthocyanins from *Hypericum perforatum* L., *Nicotiana tabacum* L., *Solanum lycopersicum* L., and *Solanum melongena* L. [33,34,35,36,37]; terpenes and steroids from *N. tabacum* L. and *Withania somnifera* (L.) Dunal [38,39]; as well as various alkaloids, including atropine, anisodine, hyoscyamine, scopolamine, anisodamine, and cuscohygrin from the HRs of *Anisodus tanguticus* (Maxim.) Pascher, *Atropa baetica* Willk, *Duboisia myoporoides* R. Br, *Hyoscyamus senecionis* Willd, and *Scopolia japonica* Maxim [40,41,42].

### 3.1. Bacterial Strains

The most frequently used *R. rhizogenes* strain for HRs induction is ATCC 15834, accounting for approximately 21% of all reported cases, followed by A4 and LBA9402, which are used in 17 and 15% of cases, respectively [43]. The HRs induced from *Swainsona galegifolia* (Andrews) R.Br. using the A4 strain exhibited a better genetic stability compared to those induced with LBA9402. The A4 strain maintained the chromosomes number of 2n = 32 in HRs, whereas HRs induced by LBA9402 showed an increased chromosome number count [43]. The choice of bacterial strain affects not only the HR induction frequency but also the capacity for SMs production. For example, the HR induction frequency (53%) and rosmarinic acid accumulation (45 mg/g dry weight (DW)) were higher with the ATCC 15834 strain compared to the A4 strain, which had a 37% induction frequency and a 19 mg/g DW rosmarinic acid accumulation [44]. The HRs from *C. acuminata* Decne induced by ATCC 15834 biosynthesized 1.0 mg/g DW of camptothecin in contrast to only 0.15 mg/g DW from HRs induced by the R-1000 strain [45]. Furthermore, HRs from *Ophiorrhiza alata* Craib induced by the TISTR 1450 strain accumulated 785 mg/g DW of camptothecin, which is twice as much as that found in the roots of the mother plant [46].

### 3.2. Nutrient Medium Optimization

Nutrient supplementation aims to enhance the SMs yield by optimizing physicochemical conditions of the culture medium. The maximum ginsenosides content (9 mg/g DW) in American ginseng (*Panax quinquefolium* C.A.Mey.) HRs was achieved in a liquid Gamborg B5 medium supplemented with 30 g/L sucrose [47]. Cultivating HRs in a lower-sucrose concentration (20–30 g/L) simulated the production of protopanaxadiol-type ginsenosides, while higher sucrose concentrations (50–70 g/L) favors the biosynthesis of Rg group ginsenosides [47]. A modified B5 medium (0.83 mM phosphate, 12.4 mM nitrate, and 0.5 mM ammonium) resulted in a maximum total ginsenosides content of 12.45 mg/g DW, which is 1.93 times higher compared to the standard B5 medium [48,49]. In HRs of *Scutellaria baicalensis* Georgi, verbascoside was the most abundant compound among martynoside and leucosceptoside A, with its maximum biosynthesis (3% DW) achieved in Murashige and Skoog (MS) medium formulated with a 50% reduction in KNO_3_, NH_4_NO_3_, and CaCl_2_ concentrations, while the concentrations of KH_2_PO_4_ and MgSO_4_ were doubled [50]. For *W. somnifera* (L.) Dunal, both the biosynthesis of withanolide A and biomass accumulation were influenced by the amount and ratio of macronutrients in the MS medium. The highest withanolide A production (15.27 mg/g DW) was observed in MS medium with double KNO_3_, whereas the highest biomass accumulation (13.11 g/L DW) occurred in MS medium with double KH_2_PO_4_. At an NH_4_^+^/NO_3_^−^ ratio of 0.00/18.80 mM, withanolide A content was 14.68 mg/g DW, whereas an NH_4_^+^/NO_3_^−^ ratio of 14.38/37.68 mM led to a biomass accumulation of 14.79 g/L DW [51].

### 3.3. Precursor Feeding

This approach is used to enhance the production of specific SMs and relies on plant in vitro systems to convert precursor compounds supplemented in the cultural medium into the desired SMs using pre-existing endogenous enzymes [52]. For instance, the addition of the precursor tyrosol (2.5 mM) was necessary to initiate salidroside biosynthesis in the HRs of *R. kirilowii* (Regel) Maxim, as tyrosol induced the activity of glucosyltransferase enzymes [53]. Supplementation of *R. kirilowii* HRs with cinnamic acid (CA) facilitated the biosynthesis of rosavin. The addition of 2.5 mM CA and 1% glucose on the 14th day of cultivation resulted in the accumulation of 0.7 mg/g DW rosarin along with 14.0 mg/g DW rosavin in biomass, while 500 mg/L of rosavin were released into the culture medium. Neither rosavin nor rosin was detected in the control samples without precursor supplementation [54]. The biosynthesis of certain betaxanthins, such as vulgaxanthin I, reached its maximum (a 3-fold increase compared to the control) in *Beta vulgaris* var. *lutea* HRs only when (*S*)-glutamate was added in the medium and sucrose was substituted by glucose [55]. Adding loganin (an iridoid glycoside) to the HR lines of *C. roseus* (L.) G.Don increased the biosynthesis of terpenoid indole alkaloids (TIAs), including catharanthine, ajmalicine, lochnericine, and tabersonine, by 26, 84, 119, and 225%, respectively [56]. The TIAs biogenesis in *C. roseus* (L.) G.Don was significantly influenced by the expression of the two key genes *DXS* (*1*-*deoxy*-*D*-*xylulose synthase*) and *G10X* (*geraniol*-*10*-*hydroxylase*) in the upper part of the TIA pathway. Inducible overexpression of *DXS* increased the production of serpentine, lochnericine, and ajmalicine by 26%, 49%, and 67%, respectively [57]. Combining genetic engineering and precursor feeding also enhanced the biosynthesis of tropane alkaloids in *Hyoscyamus reticulatus* L. HRs. The maximum content of phenols, flavonoids, and alkaloids were recorded in transgenic HRs fed with putrescine addition (4, 3, and 1 mM) after 48 h. Hyoscyamine and scopolamine content increased by 203.83% (125.09 µg/g DW) and 207.34% (161.88 µg/g DW) following 3 mM putrescine feeding for 24 h and 48 h in the control and transgenic HRs, respectively. Furthermore, the expression level of *putrescine N*-*methyltransferase* (*PMT*) and *hyoscyamine*-*6*-*beta hydroxylase* (*H6H*) genes increased 2.6- and 2.83-fold with 3 and 2 mM putrescine after 48 h of feeding in both transgenic and control HRs [58]. Simultaneous treatment with 300 µM methyl jasmonate (MeJA), together with precursors cholesterol (100 mM) and L-arginine (1 mM), improved solasodine productivity 5-fold (4.5 mg/L) in the HRs of *Solanum mammosum* L. compared to the control [59].

### 3.4. Elicitation Treatment

Elicitation is part of a plant’s defense mechanism. Elicitors are biotic (e.g., microorganisms, glycoproteins, or polysaccharides derived from plant cell walls) or abiotic (e.g., chemical: organic or inorganic compounds of non-biological origin or physical: UV or low-energy ultrasound) factors that activate signal-transduction pathways, leading to the induction or enhancement of SMs production in plant cells [60,61]. Biotic elicitors are biologically derived and can be categorized as exogenous (microbial cell walls, fungal and bacterial lysates, yeast extracts, and polysaccharides such as chitin and glucans) [62] and endogenous (cell wall polysaccharides, intracellular proteins, and small molecules such as methyl jasmonate (MeJA), jasmonic acid (JA), salicylic acid (SA), acetylsalicylic acid (ASA), pectin, chitosan (CS), coronatine (COR), and yeast extract) [63].

The process of elicitation involves many interconnected events, with mechanisms that vary according to the age of the plant material, the physicochemical environment of the plant cells, and the type, dosage, and exposure time of the elicitor. In the first step, the elicitor binds to a specific receptor on the plasma membrane, which recognizes the elicitor signal and initiates a cascade of events, including ion flux, the release of reactive oxygen species, and protein activation. In the subsequent step, the activated proteins trigger the expression of key genes involved in SMs biosynthesis pathways [64]. Examples of elicitor-induced biosynthesis of various metabolite groups, such as alkaloids, phenolic acids, terpenes, iridoids, and phenylethanoids, are provided in Table 1.

The biosynthesis of hyoscyamine increased from 8.57 to 16.78 and 19.74 mg/g DW in *Datura stramonium* L. HRs after treatment with 100 µM SA or 100 µM ASA, respectively [65]. In *Papaver orientale* L. HRs, treatment with 100 µM MeJA was more effective than treatment with 100 µM SA, increasing the production of thebaine, morphine, and codeine 2.63-, 7.18-, and 3.67-fold, respectively, compared to untreated HRs [66]. The simultaneous application of two elicitors, 100 µM MeJA and 0.5 mM β-cyclodextrin, increased the content of withaferin A (steroidal alkaloid) by more than 12-fold in *W. somnifera* (L.) Dunal HRs [67]. Conversely, exposing *W. somnifera* (L.) Dunal HRs to 150 µM SA for 4 h enhanced the biosynthesis of withaferin A, withanone, and withanolide A 42-, 46-, and 58-fold, respectively, compared to the untreated control [68].

The biosynthesis of daidzin increased 2.8- and 7.3-fold after treating *Psoralea corylifolia* L. HRs with 1 and 10 µM of JA, respectively [69]. Among the tested elicitors, including MeJA, SA, and ASA, treatment of *Astragalus membranaceus* Bunge. HRs with 283 µM MeJA increased isoflavonoid yield 9.7-fold compared to the control [70]. In *Plumbago indica* L. HRs, elicitation with 50 µM MeJA for 48 h increased plumbagin biosynthesis 1.5-fold compared with ASA treatment [71]. Elicitation with 100 µM MeJA increased the silymarin content (1.5-fold) in the HRs of *Silybum marianum* (L.) Gaertn. after 48 h of treatment [72]. In *Glycine max* (L.) Merr HRs, treatment with 100 µM MeJA for 96 h doubled the total isoflavone production yield (53.16 mg/g DW), compared to treatment with 200 µM SA (28.79 mg/g DW), which was 65-fold more than the non-elicited control [73].
ijms-25-12682-t001_Table 1Table 1Examples of elicitation as a biotechnological tool for enhanced biosynthesis of SMs in HR cultures.Metabolite(s)Plant SpeciesMetabolite Level, mg/g DWYield Increase, FoldsElicitor, µMReferenceAlkaloidsAjmalicine*Catharanthus roseus* (L.) G.Don.15.42.5MeJA, 108.85 + JA, 134.08[74]Codeine*Papaver armeniacum* (L.) DC.0.122.MeJA, 100[75]Hyoscyamine*D. stramonium* L.*D. innoxia* Mill.*D. tatula* L.11.588.8917.941.42.82.1ASA, 100[65]Indole alkaloids*Isatis tinctoria* L.3.155.9SA, 142.61[76]Morphine*Papaver armeniacum* (L.)0.154.0MeJA, 100[75]Paclitaxel*Taxus x media* var. *Hicksii*1.433.0MeJA, 100[77]Scopolamine*Anisodus luridus* Link.0.0682.0ASA, 100[78]Scopolamine*Atropa acuminata* Royle ex Lindl.10.9510.0COR, 0.5[79]Solasodine*Solanum trilobatum* L.9.331.9MeJA, 4[80]Wilforine*Tripterygium wilfordii* Hook. f.0.236.7MeJA, 50[81]TerpenesAstragaloside*Astragalus membranaceus* Bunge.5.52.1MeJA, 157.4[82]Baccatin III*Taxus x media*0.0763.0COR, 1[77]Cryptotanshinone*Salvia miltiorrhiza* Bunge.0.656.2MeJA, 100[83]Friedelin Epifriedelanol *Cannabis sativa* L.1.565.022.885.22SA, 50SA, 100[84]Ginsenosides*Panax ginseng* C.A.Mey.0.423.8MeJA, 100[85]Glycyrrhizin*Glycyrrhiza inflata* Batalin.34.795.7MeJA, 100[86]Madecassoside*Centella asiatica* (L.) Urban.1161258090COR, 1COR, 1 + MeJA, 100[87]Oleanolic acid glycosides*Calendula officinalis* L.52.523.0MeJA, 100[88]Rhinacanthin*Rhinacanthus nasutus* (L.) Kurz.6.31.7MeJA, 10[89]Tanshinone I*S. miltiorrhiza* Bunge.0.463.4MeJA, 100[83]Tanshinones*S. miltiorrhiza* Bunge.11.333.1MeJA, 100[90]Triptolide*T. wilfordii* Hook. f.0.042.0MeJA, 50[81]Triterpenoid saponins*Centella asiatica* (L.) Urban.606MeJA, 400[91]Triterpenoid saponins*Psammosilene tunicoides* W.C.Wu & C.Y.Wu.15.04.55CS, 131.01[92]Withaferin A*Withania somnifera* (L.) Dunal.13.215.6MeJA, 15[93]Withaferin A*W. somnifera* (L.) Dunal.9.5717.456.812.5MeJA, 100β-CD 5.0 mM + MeJA, 100[94]Phenolic compoundsApigenin*Ficus carica* L. cv. *Siah*0.1251.96MeJA, 300[95]Caffeic acid*Mentha spicata* L.0.161.47MeJA, 100[96]Caffeic acid*F. carica* L.0.189.0MeJA, 100[95]Caffeic acid*S. miltiorrhiza* Bunge.4.5514.06COR, 0.1[97]Chlorogenic acid*F. carica* L.6.510.5MeJA, 100[95]Chlorogenic acid*M. spicata* L.0.0151.7MeJA, 100[96]Cinanmic acid*M. spicata* L.0.0431.98MeJA, 100[96]Gallic acid*F. carica* L.0.715.9MeJA, 100[95]Salvianolic acid*S. miltiorrhiza* Bunge2.157.96COR, 0.1[97]Salvianolic acid*Salvia Przewalskii* Maxim.21.58.4MeJA, 400[98]Salvianolic acid*Salvia virgata* Jacq.2.113.76MeJA, 100[99]Rosmarinic acid*F. carica* L.1.8412.3MeJA, 200[95]Rosmarinic acid*M. spicata* L.11.840.06MeJA, 100[96]Rosmarinic acid*Prunella vulgaris* L.58.31.66MeJA, 50[100]Rosmarinic acid*S. miltiorrhiza* Bunge.8.0118.63COR, 0.1[97]Rosmarinic acid*S. przewalskii* Maxim.67.131.9MeJA, 400[98]Rosmarinic acid*Salvia virgata* Jacq.18.451.81MeJA, 100[99]Rutin*F. carica* L.0.033.34MeJA, 200[95]Quercetin*F. carica* L.0.26.91MeJA, 300[95]ASA—acetylsalicylic acid; β-CD—β-cyclodextrin; COR—coronatine; JA—jasmonic acid; MeJA—methyl jasmonate; SA—salicylic acid.


The biosynthesis of the diterpenoid andrographolide increased 8- and 5-fold after treatment with 100 µM MeJA and SA, respectively [101]. Elicitation with 100 µM JA increased the intracellular accumulation of the triterpenoid oleanolic acid 20-fold in the HRs of *Calendula officinalis* L. and enhanced its secretion in the culture medium by 113-fold [88]. The biosynthesis of cryptotanshinone and tanshinone IIA in *Salvia miltiorrhiza* Bunge HRs increased 3.9-fold, while tanshinone I and dihydrotanshinone increased 3.0- and 1.3-fold, respectively, following elicitation with 100 µM MeJA [102].

The content of harpagide, catalpol, and catalposide (iridoids), isoverbascoside, and verbascoside (phenylethanoids) was significantly enhanced in response to optimal elicitation conditions (100 µM MeJA for 72 h exposure) in 23-day-old HRs of *Rehmannia glutinosa* (Gaertn.) DC. The biosynthesis of verbascoside and isoverbascoside increased up to 10- and 6.4-fold, respectively while the harpagide content increased 7.5-fold after 72 h elicitation with 150 µM MeJA [103]. In transgenic HRs of *R. sachalinensis* Boriss expressing two putative UDP-glucosyltransferases (*UGT72B14* and *UGT74R1*), salidroside production increased by 420% and 50%, respectively, after elicitation with 300 µM MeJA, compared to the control HRs with an empty vector [104]. Tyrosine decarboxylase (TYDC) is important in salidroside biosynthesis for converting tyrosine to tyramine, while UGT is the final gene involved in salidroside biosynthesis. After SA treatment, the *RcTYDC* and *RcUGT* expressions were upregulated 49- and 36-fold compared to the control. The transgenic HR lines produced higher levels of tyramine, tyrosol, and salidroside, varying from 3.21–6.84, 1.50–2.19, and 1.27–3.47-fold, respectively, compared to their counterparts in non-transgenic lines of *Rhodiola crenulata* (Hook.f. & Thomson) H. Ohba [105]. The conversion of tyrosine into 4-hydroxyphenylpyruvate, a competing metabolite in salidroside biosynthesis, is done by *tyrosine aminotransferase* (*TAT*) enzymes. Downregulation of *RcTAT* in genetically engineered *R. crenulata* HRs led to a 6- to 60-fold increase in salidroside production [106].

### 3.5. Molecular Tools for Engineering Plant Secondary Metabolism

Metabolic engineering has been intensively utilized to enhance the biosynthesis of pharmaceutically important SMs in plant in vitro systems. These complex networks are regulated at two levels: the first level involves structural genes that encode participant enzymes of the biosynthetic pathway, while the following level of regulation is mediated by transcription factors (TFs) that control the expression of these structural genes, thereby regulating SMs biosynthesis [107].

#### 3.5.1. Metabolic Engineering

This section presents various examples demonstrating how overexpression or suppression of single or multiple genes governed by relevant TFs influence the biosynthesis of diverse groups of SMs. To better illustrate the potential of molecular tools for engineering plant secondary metabolism, we have selected a case study that explores the potential strategies for manipulating the biosynthetic pathways of morphinan alkaloids (Figure 2).

The biosynthesis of the SMs is controlled by structural genes that encode enzymes of metabolic reactions and the genes with regulatory functions that encode the TFs regulating the structural genes upon binding their promoters [108]. Several examples of metabolic engineering of SMs in HRs are summarized in Table 2.

The biosynthesis of terpenoid indole alkaloids (TIAs) was regulated by JA-responsive Apetala2/Ethylene response factor (AP2/ERF)-domain TFs such as ORCA. Overexpression of ORCA2 elevated the amount of catharanthine and vindoline in *Catharanthus roseus* HRs 2.03- and 3.67-fold, respectively, compared to the control. In contrast, the increased expression of ORCA3 activates the transcription of ZCT1 and ZCT2, which are repressors of TIA pathway [117]. Treatment with 250 µM MeJA increased the expression of ORCA TF (29–40-fold), along with TIA pathway genes like *G10H*, *strictosidine synthase* (*STR*), *tryptophan decarboxylase* (*TDC*), and *strictosidine β*-*glucosidase* (*SGD*), while the levels of ZCT remained low (2–7-fold), leading to increased concentrations of TIA metabolites (secologanin, strictosidine, and tabersonine) by 150–370% compared to untreated controls in *C. roseus* HRs. Contrarily, at a higher dosage of 1 mM MeJA, the production of TIA metabolites was inhibited, which correlated with 40-fold higher expression level of ZCT and minimal expression of ORCA (13–19-fold) and TIA biosynthetic genes (0–6 fold) [118]. An effective strategy to increase SMs production involves the “push and pull” approach through the simultaneous expression of multiple genes. Multigene engineering was employed to enhance tropane alkaloid production. Putrescine N-methyltransferase (PMT) is considered one of the first rate-limiting enzymes upstream, while tropinone reductase I (TRI) serves as a key branch-controlling enzyme, converting tropinone to tropin. Cytochrome P450 (CYP80F1) subsequently converts littorine into hyoscyamine with the final committed step catalyzed by hyoscyamine 6β-hydroxylase (H6H), which carries out consecutive oxidation reactions such as the hydroxylation of hyoscyamine and its epoxidation to scopolamine. The co-introduction of *AaPMT* and *AaTRI* in *A. acutangulus* HRs increased the content of the tropane alkaloids 8.7-fold (8.104 mg/g) compared to the control sample [41]. Similarly, the co-introduction of two key genes in camptothecin biosynthesis, *G10H* and *secologanin synthetase* (*SLS*), increased camptothecin production 2.5-fold (3.5 mg/g DW) in *Ophiorrhiza pumila* Champ. ex Benth HRs [119]. The expression levels of *PMT*, *TRI*, *CYP80F1*, and *H6H* were upregulated 14.8, 6.6, 3.1, and 11.9-fold, respectively, due to elicitation of *Anisodus luridus* HRs with 1 mM ASA. The endogenous concentrations of hyoscyamine and scopolamine increased to 57.2 and 14.7 mg/g DW, respectively, while the secretion of scopolamine in the culture medium increased 6.2-fold (1.02 mg/L) compared to the control [78].

The early and intermediate steps in the triterpenoids biosynthesis pathway involve the biosynthesis of isoprenyl diphosphate (IPP) by mevalonate-5-pyrophosphate decarboxylase (MVD) and the production of farnesyl diphosphate (FPP) utilizing IPP and dimethylallyl diphosphate by the activity of farnesyl pyrophosphate synthase (FPS). Overexpression of *PgMVD P. ginseng* HRs increased sterol concentrations (stigmasterol, sitosterol, and campesterol) up to 4.4-fold compared to the control, while ginsenoside and β-amyrin decreased. Contrariwise, overexpression of *PgFPS* increased both ginsenoside and sterol contents 2.4- and 4.6-fold, respectively [120]. The synthesis of triterpenoids such as madecassoside and asiaticoside in transgenic HRs of *C. asiatica* was slightly increased (1.15-fold) after 2 weeks of treatment with MeJA, but decreased after 4 weeks of elicitation, pointing to a feedback suppression due to *PgFPS* overexpression [121]. The DXS and geranylgeranyl diphosphate synthase (GGPPS) are key enzymes that provide the C20 parent structure for diterpenes, including tanshinones. The *SmDXS2* is a key upstream gene in the 2-C-methyl-D-erythritol-4-phosphate (MEP) pathway, while *SmGGPPS* functions downstream in the tanshinone biosynthetic pathway. Overexpression of *SmDXS2* and *SmGGPPS* together doubled tanshinone production in *Salvia miltiorrhiza* HRs to 12.93 mg/g DW, which is 21-fold higher than the control, whereas in single-gene-transformed lines, tanshinone production increased only 3-fold [122]. Single-gene-transformed lines expressing only *SmHMGR* (3-hydroxy-3-meth ylglutaryl CoA reductase) or *SmDXR* (1-deoxy-D xylulose 5-phosphate reductoisomerase) increased tanshinone production up to 3.25 mg/g DW in *S. miltiorrhiza* Bunge HRs. Co-expression of *SmHMGR* and *SmDXR* after elicitation with Ag^+^ increased tanshinone production to 6.72 mg/g DW [123].

The MYC and MYB1 TFs are positive regulators of pathway genes involved in the biosynthesis of phenolic acids [124,125]. Recently, a novel AP2/ERF TF responsive to MeJA treatment was found to stimulate the biosynthesis of phenolic acids [126]. Overexpression of SmMYB1 significantly increased the expression of *phenylalanine ammonialyase* (*PAL*), *cinnamic acid 4*-*hydroxylase* (*C4H1*), *4-hydroxyphenylpyruvate reductase* (*HPPR1*), *rosmarinic acid synthase* (*RAS1*), and *P450-dependent monooxygenase* (*CYP98A14*) in *S. miltiorrhiza* Bunge HRs, resulting in a doubling of total phenolic acid content [125]. The expression of the genes increased 1.5- and 12-fold after treatment with SA (6.9 mg/L), leading to a 1.48-fold increase (58.3 mg/g DW) in rosmarinic acid content in *Phaseolus vulgaris* L. HRs [100]. The elevated expression levels of *PAL*, *4CL*, *TAT*, *HPPR*, and *RAS* due to elicitation with 400 µM MeJA increased the content of rosmarinic and salvianolic acid B (76.1 and 21.4 mg/g DW, respectively) in the HRs of *S. przewalskii* Maxim [98]. After 6 h of exposure to 100 µM MeJA, the HRs of *Mentha spicata* L. produced 11 times more rosmarinic acid compared to the non-elicited sample, which was associated with increased expression (4.04-, 3.62-, 1.75-, and 1.45-fold) of phenylpropanoid pathway genes *PAL*, *C4H*, *4CL*, and *HPPR* [96].

#### 3.5.2. CRISPR/Cas9

The CRISPR/Cas9 is a powerful genome-editing tool widely utilized for engineering various plants or plant in vitro systems, often to validate the role of key genes involved in the biosynthesis of valuable SMs. For instance, the elimination of *OpG10H* and *OpSLS* genes by CRISPR/Cas9 in *O. pumila* HRs reduced camptothecin levels by more than 90% [119]. In *S. miltiorrhiza* Bunge HRs the transcription factor SmMYB98 positively regulates tanshinone and phenolic acid biosynthesis; knocking out this gene resulted in a decreased level of these metabolites [127]. The CRISPR/Cas9-mediated knockout of copper-containing amine oxidase (CuAO), specifically SoCuAO5 expressed in the roots, rendered *Symphytum officinale* L. HRs unable to produce pyrrolizidine alkaloids, supporting CuAO’s role in alkaloids biosynthesis [128]. N-methylputrescine oxidase (MPO) is an enzyme involved in tropane alkaloids biosynthesis. CRISPR/Cas9-mediated mutagenesis of *Atropa belladonna* L. HRs resulted in a 17–26% reduction in norhyoscyamine content, while HR lines expressing the *MPO* gene had norhyoscyamine levels 2.09- to 5.70-fold higher than the control HR line [129]. Inducing double MYC1 and MYC2 loss-of-function in tomato HRs suppressed the constitutive expression of steroidal glycoalkaloids biosynthesis genes and significantly reduced levels of α-tomatine and dehydrotomatine [130]. The silencing of the *pyrrolidine ketide synthase* gene reduced tropane alkaloids accumulation by 85% in *A*. *belladonna* HRs, indicating its crucial role in alkaloid biosynthesis [131]. Additionally, CRISPR/Cas9-based mutations of the *homospermidine synthase* (*HSS*) gene (the first pathway-specific enzyme in pyrrolizidine alkaloids biosynthesis) revealed that inactivation in only one of the two *HSS* alleles resulted in significantly reduced homospermidine and pyrrolizidine alkaloids content, whereas the HRs with both alleles inactivated, showing no detectable alkaloids [132].

In plant genome editing, CRISPR-Cas9 technology has emerged as a pivotal tool, holding the potential to revolutionize crop genetic improvement and SMs biosynthesis with its precision gene editing and regulatory capabilities [133]. Significant progress has been made in the editing of protein-coding genes. However, studies on the editing of non-coding DNA with regulatory roles are staying behind. Non-coding regulatory DNAs can be transcribed into long non-coding RNAs (lncRNAs) and miRNAs, which together with cis-regulatory elements (CREs) play crucial roles in regulating plant growth and development [134,135]. The lncRNAs are transcripts of more than 200 nucleotides in length, with minimal or no protein-coding capacity. Increasing evidence indicates that lncRNAs play important roles in the regulation of gene expression, including in the biosynthesis of secondary metabolites [135]. The triterpenoid saponin biosynthesis in *Psammosilene tunicoides* W.C.Wu and C.Y.Wu HRs was studied using SA as an efficient elicitor for SMs production. The analysis of the transcriptome-wide regulatory network revealed the identification of 430,117 circular consensus sequence reads, 16,375 unigenes, and 4678 lncRNAs. The SA upregulated the unigenes encoding SA-binding proteins and antioxidant enzymes compared to the control. The candidate transcription factors WRKY, NAC, and structural genes *AACT* (acetyl-CoA acetyltransferase), *DXS*, *SE* (squalene epoxidase), *CYP72A* might be the key regulators in SA-elicited saponin accumulation. The SA-elicitation promoted the yields of quillaic acid and gypsogenin compared to the controls with 35.4–125.9% and 20.1–28.4%, respectively. It may be speculated that the WRKY positively regulates triterpenoid saponin biosynthesis via binding to a W-box element in the promoter of *SE* [136]. In another study on *S. miltiorrhiza* Bunge, phenolic compounds biosynthesis was targeted by overexpression of a bZIP TF (bZIP2) that suppressed the production of phenolic acid in HRs [137]. Overexpression of bZIP2 in HRs resulted in a lower phenolic acid content while a CRISPR/Cas9-mediated knock-out of bZIP2 eliminated the negative regulatory effect of the bZIP TF that is believed to be able to physically interact with protein kinase 2s and cause a subsequent suppression of *PAL* gene expression, pointing to a novel TF to manipulate or enhance phenolic acids biosynthesis [137]. The overexpression of MYB36 TF in the HRs of *S. miltiorrhiza* enhanced the production of tanshinone, which was associated with a decline in phenolic acid levels [138]. Interestingly, transgene-free CRISPR/Cas9 protoplasts-originated MYB36 knockout *S. miltiorrhiza* plants resulted in white flowers, pointing to the role of these TRs as a positive regulator of anthocyanin biosynthesis [139]. The role of another bZIP TF (GbbZIP08) in promoting flavonoid biosynthesis in *Ginkgo biloba* L. was reported after its overexpression in transgenic *Nicotiana benthamiana* Domin, making it a potential candidate TF to be considered in HR transformation and gene constructs [140]. More examples of CRISPR/Cas9-mediated and other genome editing technologies for SMs’ enhancement are collected in recent review articles [141,142,143,144].

#### 3.5.3. Case Study of the Morphinan Alkaloids Biosynthetic Pathway

This case study exemplifies the challenges and opportunities associated with the metabolic engineering of the morphinan alkaloids biosynthetic pathway.

Morphinan alkaloids are benzylisoquinoline alkaloids (BIAs) that are among the most powerful narcotic analgesics for managing moderate to severe chronic pain. This class includes natural opiates like codeine and morphine as well as semi-synthetic derivatives such as dihydromorphine and hydromorphone [75,145]. Among BIAs, the most important alkaloids are morphine and codeine, papaverine, sanguinarine, and berberine [146]. The opioids antagonist naloxone and naltrexone, widely used to combat opiate addiction and overdose, are synthesized from thebaine. Thebaine not only acts as a precursor for the biosynthesis of codeine and morphine in planta but also serves as a precursor for the chemical synthesis of the analgesics like oxycodone and buprenorphine, which offer improved side-effect profiles compared to morphine [145,147]. In plants, BIAs are produced in the roots and accumulate within specialized internal secretary structures [148].

The biosynthesis of morphinan alkaloids occurs via a multistep pathway starting with the amino acid tyrosine [149]. Figure 2 illustrates the biosynthetic pathway of morphinan alkaloids starting from (*S*)-reticuline, a central intermediate in the BIA pathway. Generally, biosynthesis of BIA starts by the bioconversion of tyrosine to both dopamine and 4-hydroxyphenylacetaldehyde (4HPAA).

4-hydroxyphenylacetaldehyde is formed through L-tyrosine transamination by L-tyrosine aminotransferase (TyrAT; EC 2.6.1.5) and subsequent decarboxylation by an enzyme, provisionally named 4-hydroxyphenylpyruvate decarboxylase (4HPPDC; EC 4.1.1.80). L-tyrosine and/or tyramine undergoes *meta*-hydroxylation by an undefined enzyme (denoted as 3′OHase) yielding L-dihydroxyphenylalanine (DOPA) and dopamine, respectively. In an alternative rout L-tyrosine and DOPA are transformed to tyramine and dopamine by tyrosine decarboxylases (TYDC; EC 4.1.1.25) [150,151]. Dopamine and 4HPAA are condensed by norcoclaurine synthase (NCS; EC 4.2.1.78) via stereoselective Pictet–Spengler condensation producing (*S*)-norcoclaurine, the central precursor of BIAs [150]. Subsequently, through a series of 3′-hydroxylation, *O*- and *N*-methylations, (*S*)-norcoclaurine is then converted to coclaurine by SAM-dependent norcoclaurine 6-*O*-methyltransferase (6OMT; EC 2.1.1.128), to *N*-methylcoclaurine by coclaurine *N*-methyltransferase (CNMT; EC 2.1.1.140), to 3′-hydroxy-N-methyl coclaurine by (*S*)-N-methylcoclaurine 3′-hydroxylase (NMCH (CYP80B1), EC1.14.14.102), and ultimately to (*S*)-reticuline by 3′-hydroxy N-methylcoclaurine 4′-*O*-methyltransferase (4′ OMT, EC 2.1.1.116) [151]. (*S*)-reticuline serves as the central intermediate leading to diversification of BIAs. The biosynthesis of morphine continues with the epimerization of (*S*)-reticuline to (*R*)-reticuline. The reaction proceeds via the dehydrogenation of (*S*)-reticuline to a 1.2-dehydroreticulinium ion by 1.2-dehydroreticuline synthase (DRS; EC 1.14.19.54), which is subsequently reduced to (*R*)-reticuline by 1.2-dehydroreticuline reductase (DRR; EC 1.5.1.27) [150]. (*R*)-reticuline is then converted by the CYP salutaridine synthase (SalSyn/CYP719B1; EC 1.14.19.67) to salutaridine, which is then reduced by the short-chain salutaridine: NADPH 7-oxidoreductase (SalR; EC 1.1.1.253) into (*7S*)-salutaridinol. Subsequently, (*7S*)-salutaridinol is *O*-acetylated by acetylcoenzyme A: salutaridinol-*7*-*O*-acetyltransferase (SalAT/SAT; EC. 2.3.1.150) to salutaridinol-*7*-*O*-acetate, which is then transformed into thebaine via the formation of a C-4 and C-5 oxide bridge by thebaine synthase (THS; EC 4.2.99.24). Until the discovery of THS, this cyclization step is believed to occur spontaneously [151]. Thebaine undergoes *O*-demethylation at position six by thebaine 6-*O*-demethylase (T6ODM; EC 1.14.11.31) to yield neopinone, which spontaneously rearranges to the more stable codeinone, further reduced to codeine by cytosolic NADPH-dependent aldo-keto reductase codeinone reductase (COR; EC 1.1.1.247). Finally, codeine is converted to morphine by codeine *O*-demethylase (CODM; EC 1.14.11.32) [149]. Alternatively, the CODM initiates a minor pathway by catalyzing 3-*O*-demethylation of thebaine to oripavine. This oripavine is then transformed by T6ODM into morphinone, which is subsequently reduced by COR to yield morphine [149].

Since only plants from the genus *Papaver* can produce morphinan alkaloids, much research has focused on exploring this biosynthetic pathway in different *Papaver* species. The primary commercial natural source of morphinan alkaloids is *P. somniferum* L. (opium poppy), from the family Papaveraceae, whose medicinal properties have been recognized since ancient times [149,150]. The alkaloid content in *P. somniferum* varies between 1.5–2.7% DW of harvested material [152]. Other species investigated are *P. bracteatum* L. (Iranian or Persian poppy) and *P. orientale* (Oriental poppy). However, *P. bracteatum* produces very low amounts of codeine and morphine due to the low activity of enzymes involved in their demethylation [153], while oriental poppy produces up to 9% thebaine and 20% oripavine but does not produce morphine and codeine, making both species economically unsuitable as commercial sources for morphinan alkaloids [66]. Research has therefore focused on using plant in vitro systems and metabolic engineering for a more efficient morphinan alkaloid production. Hairy root cultures for the production of BIAs have been initiated from *P. somniferum* [154], *P. bracteatum* [148,155], *P. orientale* [66], *P. armenicum* [75], and *Macleaya cordata* (Willd.) R.Br. [156]. Considerable efforts have been made to optimize HR cultures to enhance morphinan alkaloids biosynthesis. In *P. bracteatum* hairy roots, overexpression of *SalAT* increased the levels of thebaine, codeine, and morphine by 1.28, 0.02, and 0.03%, respectively, compared to the control [157]. Overexpression of *COR* in the HRs of the same species increased the codeine and morphine content by 0.04% and 0.28%, respectively, compared to the control [153]. In *P. orientale* HRs the MeJA or SA induced overexpression of *COR*, *SalAT*, *SalR*, *T6ODM*, *CODM*, and *SalSyn*, resulting in enhanced thebaine (up to 2.63-fold) and morphine (up to 6.18-fold) production compared to the control [66]. In *P. armeniacum*, elicitation with 100 µM MeJA increased the expression of *TYDC*, *SalAT*, *T6ODM*, and *COR*, leading to increased biosynthesis of thebaine (41 µg/g DW), codeine (120 µg/g DW), and morphine (150.67 µg/g DW) [75]. 

## 4. Bioreactor Technology for Hairy Roots Cultivation

The demand for plant-derived SMs is increasing exponentially, due to their diverse health-promoting properties. To meet this growing demand in the global market, it is necessary to produce these SMs on a large scale, utilizing bioreactor technology as the best alternative for scaling up the production of SMs from plant in vitro systems [158]. However, when the in vitro cultures are transferred from shake flasks to bioreactors and scaled-up from pilot to industrial scale, the cultivation environment of the plant cells, especially HRs, can change in terms of hydrodynamic shear forces and rheological properties. This can lead to altered effects of shear stress, oxygen supply, and gas composition, which may reduce productivity in terms of biomass and SMs [158]. A major bottleneck that hampers the industrial utilization of HRs is their complex morphology and non-uniform growth pattern [28]. The tangled and fibrous-clump nature of HRs results in the formation of an interlocked network, external boundary layers over the root surfaces, and stagnant zones within root clumps. This creates a non-homologous culture environment impeding mass transfer for the delivery of nutrients and oxygen, and leads to the formation of senescent tissues. Oxygen deficiency and gradients, resulting in decreased dissolved oxygen concentrations, are key challenges in the mass production of HRs and are considered a growth-limiting factor in HR bioreactors [28,159]. Additionally, HRs are delicate and fragile; while vigorous mixing can augment mass transfer, it also increases hydrodynamic shear stress, which reduces the vitality and productivity of the HRs due to callus formation [160]. Table 3 presents the main morphological characteristics of HRs that distinguish them from plant cell suspensions, microbial, and animal cells.

Therefore, scaling up the HRs cultivation is not straightforward and requires careful consideration of factors such as oxygen supply (dissolved oxygen, dO_2_) and CO_2_ exchange, pH, medium composition, agitation, aeration, and cell density [165,166]. Continuous O_2_ supplementation is needed to support the biological activates of the growing HRs, while agitation ensures the homogeneity of the liquid medium. The aeration system typically consists of stainless-steel sparger, with an agitator or impellers required for mass/heat transfer and uniform air distribution. Another important consideration is the high shear and hydrodynamic stress that are experienced by HR cultures. Shear stress is caused by continuous agitation and aeration of the medium, as well as by distribution and fragmentation of the gas bubbles and bubble rupture at the liquid surface. Currently, there is no universal strategy to minimize shear stress, which is thought to be influenced by certain factors such as tissue morphology, culture age, aeration and agitation speed, and the viscosity of liquid medium [28]. Predominantly, HRs cultivation is performed at low agitation speeds between 75–150 rpm, with 10–30% dO_2_ (corresponding to 0.1–0.5 vvm airflow) controlled by sparger or 0.5–1.0 L/min using an airflow meter [165]. An essential parameter that provides information about the oxygen availability for HRs in liquid cultures is its transfer coefficient (kLa values), which represents the fraction dissolved in water. The reduced oxygen level is a function of the metabolic activity of the growing biomass and culture yield. HRs have a lower metabolic activity and doubling time than microbial cells, requiring a lower O_2_ supply. The level of dO_2_ in bioreactor liquid cultures can be adjusted by agitation or stirring, aeration, gas flow, and air bubble size. Temperature is another critical and controlled parameter, which is usually standardly maintained at 25–26 ˚C [165].

### 4.1. Bioreactor Design and Process Intensification

Considering the unique challenges of HRs cultivation, the prime focus when designing a suitable bioreactor is on achieving adequate mixing of the culture medium with minimized shear stress and optimized mass transfer while reducing the hydrodynamic pressure [28]. A bioreactor for HR cultivation must meet several key requirements upon scale-up: (1) its geometric and flow characteristic must remain effective; (2) it must provide nutrients and oxygen in sufficient concentrations and make them accessible to the roots; and (3) it should facilitate the scale-up process by having identifiable and quantifiable process parameters that can be replicated at larger scales efficiently and cost-effectively [160]. The key issue in bioreactor design and operation is to control the biochemical processes consistently and optimally for maximum productivity. This optimization can be approached on two levels: (1) the biological entity and its products, including metabolite synthesis and accumulation; and (2) physical parameters, such as oxygen supply, temperature, medium continuity, and product removal. The second level basically involves designing the reactor vessel in a way that maximizes physical parameters, maintaining them in equilibrium to achieve consistent productivity [28].

On that account, various bioreactor configurations have been employed for HR cultivation, generally classified into liquid-phase, gas-phase, and hybrid systems (a combination of both) [28]. The general advantages and disadvantages of each bioreactor type are summarized in Table 4. In liquid-phase reactors, HR biomass is fully submerged in the liquid medium. Given that mass transfer of gaseous media and high shear stress are the key limiting parameters in these bioreactors, many different vessel designs have been developed to address these challenges. The most frequently used bioreactors for HRs cultivation are presented in Figure 3.

Among the most frequently used bioreactors for HR cultivation are the mechanically driven stirred tank reactors (STRs), pneumatically driven bubble column (BCRs), airlift reactors (ALRs), and convective flow reactor (CFRs) [159]. STRs are conventional bioreactors known for their characteristic mass and heat transfer properties. They consist of impeller or turbine blades and a sparger that regulate aeration, bubble size, and medium currency [168]. Although the STRs provide excellent control of temperature, dO_2_, and nutrient concentration, the impellers cause higher shear stress on the roots than other bioreactor types. The specific morphology of the HRs often leads to damage from the rotating impellers, resulting in wounding, callus formation, and a negative effect on biomass accumulation and SMs biosynthesis [168]. Therefore, HR cultivation in STRs should be conducted with restricted impeller input and speed, and several adaptations, such as impeller modifications or the use of steel cage or perforated mesh to separate HRs from the impeller, may be employed [169].

Bubble column reactor consist of a vertical column with a sparger at the bottom facilitating the release of air or a gas mixture for both aeration and mixing. These reactors are easy to scale up, have low capital and operational costs, and significantly minimize HR shear stress due to the use of bubbles. However, the undefined flow patterns and non-uniform mixing in high-density cultures can lead to reduced growth performance, often due to a decrease in gas-liquid interface areas. Introducing multiple spargers, dividing the bubble column into segments, or increasing the aeration rate alongside HR growth and density, may mitigate these disadvantages [28,159]. ALRs have a similar construction to BCRs but include an external or internal draft tube. This tube separates upward and downward flows, providing liquid circulation by directing airflow through a sparger at the bottom for gas exchange. The density difference between the flows facilitates liquid circulation, prevents bubbles coalescence, and enhances oxygen mass transfer due to the increased number of bubbles [167]. The turbine blade reactor unites features of STRs and ALRs, where the cultivation space is separated from the agitation one by a stainless-steel mesh, significantly reducing the shear stress by preventing HR contact with the impeller. Air is delivered through the bottom of the reactor and dispersed by an eight-blade stirrer impeller [169]. The spin filter reactor (SFR) uses a rotating filter to the cultures and allows simultaneous removal of the spent medium and the addition of a fresh one [169]. Spin filter reactor consist of an STR with an external culture chamber for HR growth, and the medium circulates through both chambers. Although this bioreactor type may offer improved performance, it is difficult to scale up due to the high pressure required to circulate the culture medium [28,159]. The rotating drum reactor (RDR) operates similarly to a fill-and-drain reactor. It consists of a drum-shaped container placed on rollers providing support and rotation. The drum rotates at slow speed (2–6 rpm) to minimize the shear pressure on the HRs, and a polyurethane foam sheet onto the drum surface is used to attach (immobilize) the HRs without breaking. The rotating motion facilitates proper gas-liquid mixing and promotes efficient oxygen transfer to biomass at high densities [28,159]. Another cultivation system that uses repeating cycles of liquid and gas phases is the ebb and flow reactor (EFBR). It is designed for repetitive cycles of filling and draining between the reactor vessel and medium reservoir [28]. This bioreactor type offers improved mixing and mass transfer capacity, including effective manipulation of the gas composition, such as the addition or removal of O_2_, CO_2_, or ethylene [160]. Due to its advantages, such as limited shear damage, reduced hypertension (pressure stress), facilitated medium change, and improved balance between oxygen and nutrients access, temporary immersions systems (TIS) have been introduced for HR cultivation. HRs are placed on the netting of the cultivation chamber, while the nutrient tank is located at the bottom of the vessel. These systems are based on the concept of temporary contact between the cultured tissue and liquid medium, and optimizing the frequency and time of immersion can enhance HRs growth and SMs production. One of the most frequently used TIS is the RITA^®^ (Recipient for Automated Temporary Immersion System) from the French company Vitropic [170].

In gas-phase reactors, HRs are fixed (immobilized) in the culture vessel on horizontal sheets or rings of inert material. HRs are exposed to a mixture of ambient air and nutrient medium, which is dispersed or trickled over the HR bed in a spray, mist, or droplet mode, while the unused medium is collected and recirculated [28]. Cultivation in gas-phase reactors comprises a two-stage cultivation system, where the first stage is a liquid-phase reactor regime, allowing HRs to disperse and grow on the support matrix [17]. Gas-phase bioreactors solve key challenges associated with liquid-phase reactors, such as oxygen mass transfer, and provide a lower shear stress environment and complete control of gases in the culture environment [159]. However, a major disadvantage is the channeling of liquid through the bed, often forming a viscous liquid film coating the roots, creating a high mass transfer barrier [17]. Some of the most frequently used gas-phase reactors include nutrient mist (NMRs), trickle bed (TBRs), droplet reactors, and radial flow (RFR) reactors [159]. In NMRs, the HRs are immobilized in a growth chamber, and the culture medium is delivered from the top over the root bed as an aerosol (nutrient mist) produced by ultrasonic methods, nozzles, or compressed air where spent medium is drained from the bottom to a reservoir and recirculated. The droplet sizes in NMRs are usually 0.01–10 μm, which decrease the thickness of the liquid film deposition on the root tissue surface, facilitating nutrient and gas exchange. NMRs offer unique advantages, including low pressure drops and shear rates, high nutrient transfer rates (because of the large mass transfer area), rapid replenishment of nutrients, removal of toxic metabolites, easy control of gas composition, operation, and scale-up [28,159,167]. In contrast, TBRs have a larger droplet size (over 10 μm), resulting in a thicker liquid film on the plant tissue surface, which hinders gas transfer. The tendency for the root bed to accumulate liquid and the absence of agitation are major limitations for large-scale design of TBR [167]. RFRs use a specific cross-sectional area where the air-saturated medium enters a sidewall in the reactor and exits through ports at the center of the top and bottom plates. This design improves the oxygen supply and is suitable for high-density HR culture [159].

Hybrid bioreactors for HR cultivation aim to overcome some of the main disadvantages observed in liquid-phase reactors such as the formation of stagnant zones with inadequate nutrient transfer and oxygen depletion, and gas-phase reactors like the manual distribution of the growth chamber. In hybrid bioreactors, during the initial cultivation stage, the reactor operates in a liquid-phase regime allowing the roots to attach uniformly to the anchoring system before switching to the gas-phase reactor for more effective HR cultivation. Well-accepted hybrid reactors include combinations BCR and TBR, and BCR and NMR [28]. 

Hairy root cultures have been successively cultivated in STRs or modified STRs for the biosynthesis of various secondary metabolites, achieving concentrations of SMs and growth characteristics similar to or superior to those obtained in shake flasks demonstrating their suitability for HR cultivation [171,172]. For example, the cultivation of *C. roseus* (L.) G. Don HRs in a 5-L STR (4 L/min flow rate equivalent to 40% dO_2_, marine-type impeller working at 100–120 rpm, 25 °C, and illumination of 300 lux) resulted in an 11-fold increase in biomass and a 2-fold increase in ajmalicine content (0.029 mg/g DW) [172]. Several modification of STRs, such as the use of perforated Teflon disk [173,174], nylon, or stainless-steel mesh supports, have been employed [171]. Examples illustrating HR cultivation and SMs biosynthesis in different bioreactor configurations, including modifications and cultivation parameters, are presented in Table 5. The HR cultivation of the medicinal herb *Boerhaavia diffusa* L. in a 10-L STR with a nylon mesh, which facilitated high-density HR cultivation and reduced the shear stress, resulted in 6.1- and 2.1-fold increases in boeravinone B and eupalitin accumulation, respectively, compared to the shake flask cultivation, without negatively effecting biomass yield [171]. HR growth has been protected from the shear forces of the impellers in a STR by the addition of perforated Teflon disk. Batch cultivation of *A. annua* L. HRs in this setup increased biomass accumulation 1.5-fold, while artemisinin production increased 1.4-fold [173]. However, using the same bioreactor configuration in a fed-batch mode, artemisinin production increased almost 4.0-fold compared to the batch mode of cultivation [174].

Another strategy to enhance SMs production involves elicitation and precursor feeding. For instance, the application of 10 mg/L MeJA improved artemisinin content 2.4-fold during batch cultivation in an STR, compared to the non-elicited cultivation [173]. In a similar manner, the addition of appropriate elicitors and precursors, although to a lesser extent, increased the total alkaloid production in *Vinca minor* L. HRs, cultivated in a 5-L STR [178]. Metabolic engineering aimed at increasing the expression of key genes in the curcumin biosynthetic pathway in *A. belladonna* HRs resulted in a 2.3-fold improvement of curcumin yield in a modified STR, compared to the shake-flask cultivation [176].

Bubble column reactors (BCRs) bioreactors are also suitable for HR cultivation, providing sufficient oxygen and nutrients. A balloon-type BCR was effective for scaling up *P. ginseng* C.A. Mey. HR cultivation from 5 to 20-L volumes [196]. Scaling up the cultivation of *Clitoria ternatea* L. HRs in BCRs resulted in a 4-fold increase in total alkaloid content (34.8 mg/g DW) in comparison to the shake-flask cultivation [197]. Elicitation with 15 mg/L pullulan at the late exponential growth phase during *B. vulgaris* L. HR cultivation in a BCR with a plastic basket for HR anchoring resulted in 47% higher betalaine production compared to the non-elicited control, whereas betalaine content was even lower in the shake-flask experiment [186]. Likewise, using MeJA increased betalaine content 1.4-fold (36.13 mg/g DW) during *B. vulgaris* L. HR cultivation in a 3-L BCR. Simultaneous feeding with spermidine and putrescine, both at concentrations of 0.75 mM, resulted in a similar increase of 1.27-fold in betalaine content (32.9 mg/g DW), compared to the control [187]. 

The importance of bioreactor modifications, especially the supportive matrices for HR anchoring, was highlighted by comparing azadirachtin biosynthesis in *A. indica* L. HRs cultivated in conventional STRs and BCRs, and modified BCRs with a polypropylene mesh or foam as root support. In conventional STRs and BCRs, HR growth was not significant compared to the initial inoculum, and azadirachtin production was not reported likely due to high shear stress on the HRs, as indicated by elevated phenolic levels. In the modified BCR, biomass accumulation was 9.5 g/L DW with polypropylene mesh and 9.2 g/L DW with polyurethane foam. However, azadirachtin content was higher in BCR with polyurethane foam (28.52 mg/L) than with polypropylene mesh (20.23 mg/L) due to polyurethane foam’s superior porosity and absorbency, which more effectively facilitates oxygen and mass transfer [183]. 

Similarly, *C. roseus* (L.) G. Don HR biomass accumulation and ajmalicine biosynthesis were highest in a BCR with polyurethane foam. Biomass accumulation in this bioreactor was 2.33-, 4.53-, and 1.22-fold higher than in the conventional BCR, rotating drum reactor (RDR), and modified BCR with polypropylene mesh, respectively. Ajmalicine content was 2.19-, 7.39-, and 1.13-fold higher in BCR with polyurethane foam compared to the other bioreactor types [188]. ALR have been proven to be suitable for HR growth [198] and have also been integrated for the production of diverse SMs, including phenolic acids [192], flavonoids [199], terpenes [194], and even human growth hormone (hGH1) [200]. The biosynthesis of puerarin from the HRs of *Pueraria phaseoloides* (Roxb.) Benth was approximately 200-fold higher (5.59 mg/g DW) when cultivated in 2.5-L disposable ALR than in shake flasks [199]. Cone ALR produced the highest levels of betacyanin (27 mg/g DW) and biomass (6 g/L) from *B. vulgaris* L. HRs compared to other bioreactor constructions, such as balloon, bulb, drum, and column reactors [195]. The insertion of stainless-steel mesh in ALR is a common modification that positively affects the HR growth and SMs biosynthesis. The production of hGH1 increased 1.53-fold when the HRs of *Brassica oleracea* var. *italica* were grown in an ALR with a mesh compared to flask cultivation [200]. The biosynthesis of sotolone and 3-amino-4,5-dimethyl-2(5H)-furanone increased by up to 1.7- and 17% of the volatile fraction in *Trigonella foenum*-*graecum* L. HRs cultivated in an ALR with a mesh [201]. Gas-phase reactors, such as nutrient mist reactors (NMR), provide better growth and biosynthetic characteristics than liquid-phase reactors. For example, artemisinin production from *A. annua* L. HRs was 3-fold higher than in BCR [182]. Acoustic NMR showed the best performance in terms of increased specific growth rate and esculin content, which was twice as high as in the bubble-column and nutrient-sprinkle reactor. This effect was attributed to the growth of the HRs in acoustic NMR on mesh anchorage, which supports high mass transfer with increased absorption rates in the dispersed phase [189]. Thiophene (a sulfur-containing aromatic compound) content was also twice higher in *Tagetes patula* L. HRs cultivated in acoustic NMR compared to nutrient-sprinkle reactors [202]. Due to the nano-size nutrient mist particles generated in the NMRs, which promote uniform growth during high-density cultivation of *A. indica* L. HRs, azadirachtin production was 2.5-fold higher than in BCR [177]. Artemisinin production from *A. annua* L. HRs was nearly 5-fold higher in NMR than the BCR [190]. The production of mouse interleukin-12 was 5.3 µg/g FW in NMR, which was 49.5% higher compared to the ALR [203].

The advantages of the TIS concept also appear beneficial for HR cultivation. The cultivation of *Centaurium maritimum* (L.) Fritch in RITA^®^ bioreactors resulted in up to a 4-fold increase in biomass accumulation and an 8-fold higher total secoiridoid glycosides production compared to shake-flask cultivation [204]. Similarly, cultivation of *Gentiana dinarica* Beck HRs in RITA^®^ bioreactors yielded the highest xanthones production (56.82 mg/vessel), compared to 18.08 mg/vessel in a BCR system [205]. RITA^®^ systems achieved the highest biomass accumulation of *Agastache rugosa* (Fisch. & C.A.Mey.) Kuntze HRs. Although the rosmarinic acid content (4.49 mg/g DW) was half that in the nutrient-sprinkle bioreactor, it was comparable to the content (4.65 mg/g DW) achieved in shake flasks [206].

Driven by the market demand and the need for cost minimization, disposable/single-use bioreactors (SUB) have gained significant attention in plant cell-based processes over the past few decades [207]. Unlike traditional glass or stainless-steel bioreactors, SUBs cultivation containers are made of FDA-approved disposable biocompatible plastics (polyethylene, polystyrene, polytetrafluorethylene, or polypropylene). The pre-sterile cultivation containers remove the necessity for cleaning and sterilization and are discarded after product harvest. While challenges such as proper mixing, mass transfer, and process measurement and control may arise, these vessels are equipped with necessary miniaturized sensors, sampling ports, and piping sections. Therefore, SUBs offer high flexibility in terms of time and costs, typically reducing costs by 30–40% [207,208,209]. Currently, the most popular SUBs include STR, NMR, and pneumatically driven SUBs, which are similar in principle to conventional reactors; orbitally shaken SUBs, which provide mixing through orbital oscillations of the cylindrically shaped vessel; and rocking SUBs (wave-mixed bioreactors), which use an agitation mechanism originating from the rocking of a platform with a disposable bag-like container [203,207]. For HRs, the most used SUBs are wave-mixed and NMRs [203,210,211,212]. 

Wave-mixed bioreactors consist of a disposable bag containing cells and media mounted on the bioreactor platform. Headspace aeration inflates the cell bag, and the rocking motion of the platform (which can be one, two, or three-dimensional) provides mixing and mass transfer. The dO_2_ and pH are typically measured by optical probes, and gas blending is supplied using mass flow and integrated controllers [209]. The wave-mixed bioreactor system provided a suitable environment for the growth and SMs biosynthesis of *Rindera graeca* (A.DC.) Boiss. & Heldr HRs. Root growth was 2-fold higher than in shake-flask cultivation, while deoxyshikonin production was observed only in the wave-mixed bioreactor [211]. The wave bioreactor was the most efficient in promoting HR growth of *P. ginseng* C.A. Mey. compared to conventional spray reactor and shake flask cultivation, with ginsenoside content reaching 145.6 mg/L, almost 3-fold higher than that obtained during shake-flask culturing [210]. The disposable NMR was also suitable for the HR cultivation of *A. annua* L. and *A. hypogaea*, even when scaled from 1- to 20-L, resulting in a biomass accumulation almost 1.2-fold higher than in shake flasks [212]. Similarly, the disposable NMR exhibited better growth and biosynthetic characteristics than the conventional ALR when total production of the mouse IL-12 from tobacco HRs was approximately 50% higher [203].

### 4.2. Hairy Roots Multiple Applications, Recent Patents, and Commercial Examples

The combined efforts in basic and applied research focusing on initiation and selection of fast-growing and highly productive HR lines, nutrient medium optimization, increased knowledge of cellular processes and factors regulating SMs biosynthetic pathways, and the development of suitable bioreactors with optimized physical parameters, have paved new avenues for designing highly efficient HRs for the biosynthesis of valuable SMs and effectively operating bioreactor systems, leading to novel technologies subject to patenting [170]. The patents cover the production of various SMs (with potential application in cosmetics, dietary supplements, and drugs) from HR cultures as well as the technical solutions for large-scale HR cultivation through modifications of existing bioreactors or the design of a new ones. Several technologies for SM biosynthesis and isolation from HRs have been described, including the production of four new flavonoid glycosides and two new sesqueterpene glycosides from *C. roseus* (L.) G. Don HRs [213], the anti-diabetic compound serpentine from the same plant species [214], betanin production from *B. vulgaris* L. HRs [215], the neurotransmitter precursor L-DOPA, used to relieve Parkinson’s disease symptoms biosynthesized by *B. vulgaris* L. HRs [216], and the triterpenoid sapogenin from *Medicago truncatula* Gaertn HRs, which, when elicited with 25 µM β-cyclodextrine, increased the biosynthesis of erythrodiol and oleanolic acid by 150- and 2-fold, respectively [217]. The combined use of elicitors and suitable TIS for the biosynthesis of stilbenes and stilbene derivatives from *Arachis hypogaea* L. HRs has been reported. Elicitation of the HRs with 10.2 mM sodium acetate during cultivation in an intermittent immersion vessel resulted in a 60-fold increase in *trans*-resveratrol compared to the non-elicited control [218]. The same cultivation system was used for ginsenosides production from *P. ginseng* C.A. Mey. HRs [219]. 

The suitability of the 15, 250, 350, and 1000-L bubble-column bioreactor for HR cultivation [220], the betalaines biosynthesis from *B. vulgaris* L. in a 3-L BCR [221], and the production of the anti-inflammatory compound bisabolol from *Matricaria recutita* L HRs in conventional mechanically and pneumatically driven bioreactors were reported [222]. Alongside conventional bioreactors, the cultivation of *Taxus chinensis* (Rehder and E.H. Wilson) Rehder in 5- to 300-L external circulation NMRs has been patented [223]. A flexible wall bioreactor using a small droplet-size mist unit and a flexible culture chamber was found suitable for increasing HR growth and density [224]. In another patent, an NMR with a rotatable culture bed provided optimal condition for HR growth [225], even when scaled up to 1000-L [226].

Despite the potential and patents for HR cultivation and production, sustainable industrial cultivation remained unexplored. The only company producing a wide range of products for cosmetics and nutrition from HRs was RooTec, Switzerland. Their portfolio included the production of atropine, gingsenosides, coumarines, flavonoids, alkaloids, camptothechin, anabasine, and nicotine from HRs of *Atropa belladonna*, *Carlina acaulis*, *Nicotiana glauca*, and *Panax gingsen* [29,166]. The Swiss Company Mibelle Biochemistry Mibelle AG developed RootBioTec™ HW, a product based on basil hairy root extract designed to promote fuller and denser hair [227]. *Ocimum basilicum* L. HR extract was incorporated in the Sunumbra^®^ sport natural sunscreen, produced by Organic Products LLC, USA [228]. A concentrated mixture of cell wall fraction and hydro-ethanolic extract, both derived from *Brassica rapa* HRs, was used in Vita Genesis White product of Vitalab s.r.l., Italy, a product dedicated to safely inhibiting pigmentation, reducing melanin synthesis, and regenerating the extracellular matrix for a luminous and even skin tone [229]. Researchers from Samabriva, SA Company, France, investigated the capacity of *B. rapa* HRs for producing the recombinant protein alpha-L-iduronidase (IDUA), clinically important as an enzyme-replacement pharmaceutical for treating mucopolysaccharidosis type I (MPS I), a progressive lysosomal storage disorder. Due to the optimized secretion medium, IDUA activity increased more than 150-fold compared to non-induced medium [230]. The *B. rapa* HR-based expression system was successfully transferred to a pilot-scale 25-L airlift bioreactor, demonstrating that recombinant protein IDUA has highly homogeneous posttranslational profiles, ensuring strong batch-to-batch reproducibility [231]. Funded in France in 2011 with an R&D center focused on developing bioproduction processes for customers, Samabriva, SA, established a biomanufacturing unit in Belgium in 2023 aimed at industrial-scale production of high-value molecules. The company is now focused on industrializing HR-based expression systems for the sustainable production of natural products, including recombinant proteins in proprietary bioreactors of at least 350-L [232].

## 5. Conclusions and Future Perspectives

For decades, plant in vitro systems have offered a suitable alternative to whole plants for continuous production of safe, superior-in-yield, and high-quality natural products at a lower cost. Although chromosomal changes and fluctuating yields of SMs are reported in plant in vitro systems (e.g., cell suspensions), HRs, in particular, are genetically stable and offer a biosynthetically consistent platform with a high growth and productivity rate. Therefore, HRs are gaining increasing attention as a bioproduction system and are now considered a valuable approach for both basic and applied research on plant-derived natural products biosynthesis. When developing commercial processes for SMs biosynthesis based on HRs, several factors regarding sustained production and efficiency must be considered. Strategies like nutrient medium optimization, precursor feeding, elicitation, and metabolic engineering (e.g., editing of non-coding regulatory DNA/RNA sequences, overexpression of biosynthetic genes and TFs, or suppression of catabolic or competing pathway genes) have been employed to boost the biosynthesis of these low-volume, high-value SMs. Recent advancements in genomic engineering, particularly CRISPR/Cas9 technology, open new possibilities for modifying metabolic pathways and creating custom, high-yielding HR lines capable of producing functional molecules for application in food, cosmetic, and pharmaceutical industries. The combination of CRISPR/Cas technology and non-coding regulatory DNA has great potential to generate novel alleles that affect various agronomic traits of crops or medicinal plants for precise regulation of the biosynthesis of SMs. However, commercial exploitation of these SMs requires scaling up of the HRs in bioreactors. Considering the unique morphological peculiarities and sensitivity of HRs, the suitable bioreactor design must provide optimized oxygen transfer, adequate growth conditions, appropriate agitation, aeration, a homogeneous culture environment, and a minimized shear stress. Various liquid-phase, gas-phase, and hybrid bioreactors with numerous modifications have been designed to meet these challenges.

The major bottleneck that hampers the industrial production of SMs based on HRs cultivation is the poor understanding of the SMs metabolic pathways and their regulation, which further reflects on their competitive economic value. Therefore, the future focus should be on the integration of different approaches, including omics (genomics, transcriptomics, proteomics, and metabolomics) technologies to understand metabolite synthesis pathways through the identification of missing/unknown metabolite pathway enzymes or to identify enzyme variants with higher activity that are expected to obtain higher productivity of SMs. Along with that, the initial investments and overall operating costs during the cultivation process are correlated with the appropriate design of the bioreactor and the optimized process parameters. The use of disposable bioreactors is the preferred choice for the reduction of the production costs, while the application of mathematical models and real-time monitoring can be harnessed to optimize the cell-cultivation process and enhance the productivity in plant cell bioreactors at a commercial scale.

## Figures and Tables

**Figure 1 ijms-25-12682-f001:**
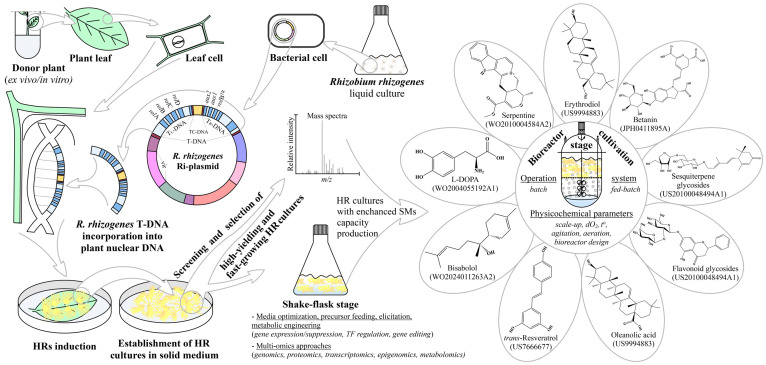
Schematic presentation of the HRs induction process, biotechnological toolkit used for SMs enhancement, HRs multiple applications, and patents related to the topic. Sterile leaf explants are exposed to *R. rhizogenes* culture for transfer and integration of T-DNA from the bacterium Ri-plasmid to the plant genome. After HRs induction, individual lines are established and the selection of high-producing and fast-growing clones is performed. Root cultures with higher production capacity after the shake-flask stage are transferred into bioreactors with suitable design for scaling up the production process, which can finally be patented. During the shake-flask and bioreactor stages, different strategies for the enhancement of SMs biosynthesis, as well as process parameter optimizations, are applied.

**Figure 2 ijms-25-12682-f002:**
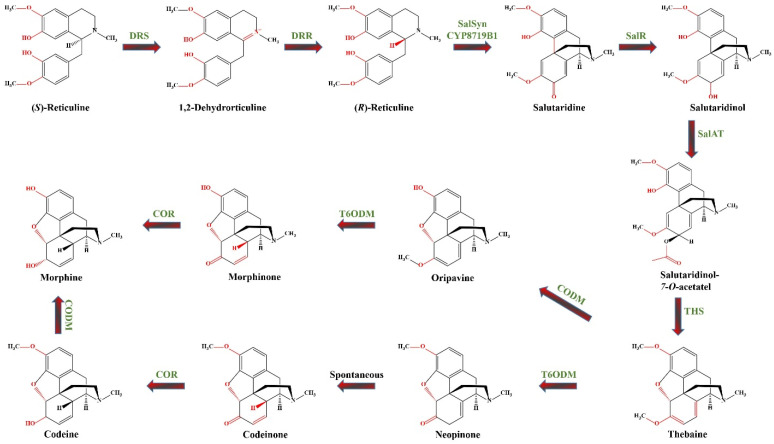
The metabolic pathway of morphinan alkaloids biosynthesis, showing two routes from thebaine to morphine. DRS—1.2-dehydroreticuline synthase (EC 1.14.19.54); DRR—1.2-dehydroreticuline reductase (EC 1.5.1.27); SalSyn/CYP719B1—CYP salutaridine synthase (EC 1.14.19.67); SalR—salutaridine: NADPH 7-oxidoreductase (EC 1.1.1.253); SalAT/SAT—acetylcoenzyme A: salutaridinol-*7*-*O*-acetyltransferase (EC. 2.3.1.150); THS—thebaine synthase (EC 4.2.99.24); T6ODM—thebaine 6-*O*-demethylase (EC 1.14.11.31); COR—codeinone reductase (EC 1.1.1.247); CODM—codeine *O*-demethylase (EC 1.14.11.32).

**Figure 3 ijms-25-12682-f003:**
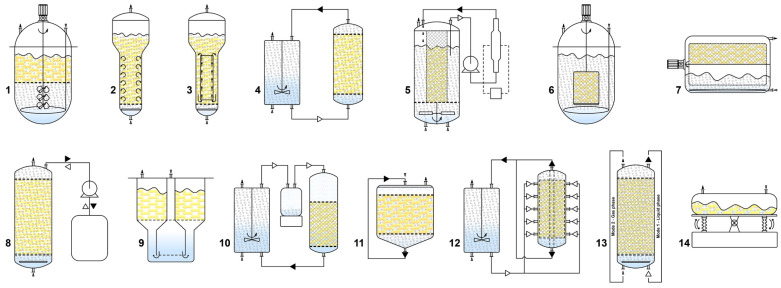
Various types of liquid-phase (1—modified stirred tank; 2—bubble column; 3—airlift; 4—convective flow; 5—turbine blade; 6—spin filter; 7—rotating drum; 8—ebb and flow), 9—temporary immersion system (RITA^®^ bioreactor), gas-phase (10—nutrient mist; 11—trickle bed; 12—radial flow), hybrid (13—hybrid reactor, working as a liquid-phase reactor in mode 1 and a gas-phase reactor in mode 2), and single-use (14—wave-mixed) bioreactors, used for upscaling of hairy root cultures.

**Table 2 ijms-25-12682-t002:** Examples of metabolic engineering strategies to enhance the SMs biosynthesis in HRs.

Metabolite(s)	Plant Species	Metabolite Level, mg/g DW	Yield Increase: Folds	Over Expressed Genes/TFs	Reference
Alkaloids					
Camptothecin	*Camptotheca accuminata* Decne.	1.12	1.5	ORCA3	[109]
Catharanthine	*Catharanthus roseus* (L.) G.Don.	1.96	6.5	*G10H*, ORCA3	[110]
Hyoscyamine	*Anisodus tanguticus* (Maxim.) Pascher.	1.9	2.3	*TRI*, *PMT*	[41]
Hyoscyamine	*Hyoscyamus niger* L.	0.2	1.25	*PMT*, *H6H*	[111]
Hyoscyamine	*H. reticulatus* L.	0.13	3.0	*PMT*, *H6H36*	[58]
Scopolamine	*Datura innoxia* Mill.	0.54	2.16	*H6H*	[112]
Scopolamine	*H. niger* L.	0.31	1.20	*PMT*, *H6H*	[111]
Scopolamine	*H. reticulatus* L.	0.16	2.6	*PMT*, *H6H*	[58]
Terpenes					
Platycosides	*Platycodon grandiflorum A.* DC.	1.6	2.5	*PgHMGR*	[113]
Tanshinones	*Salvia miltiorrhiza* Bunge.	2.73	4.74	*SmHMGR*/*SmGGPPS*	[114]
Withaferin A	*Withania somnifera* (L.) Dunal	17.45	12.5	*HMGR*, *SQS*, *SMT*-*1*, *SDS*/*CYP710A*	[94]
Phenolic compounds					
Apigenin	*Ficus carica* L.	0.16	51.96	*CHS*, *F3′H*, *PAL*, *UFGT*, MYB3, bHLH	[95]
Caffeic acid	*Leonurus sibirica* L.	11.4	2.7	AtPAP1	[115]
Chlorogenic acid	*L. sibirica* L.	19.4	4.75	AtPAP1	[115]
Gallic acid	*F. carica* L.	0.71	5.9	*CHS*, *F3′H*, *PAL*, *UFGT*, MYB3, bHLH	[95]
Total phenolics	*S. viridis* L.	38.65	1.3	*TAT*, *HPPR*, *PAL*, *C4H*, *4CL*, *RAS*	[116]

*G10H*—geraniol-10-hydroxylase; *TRI*—tropinone reductase I; *PMT*—putrescine N-methyltransferase; *H6H*—hyoscyamine 6β-hydroxylase; *HMGR*—3-hydroxy-3-methylglutaryl-coenzyme A reductase; *GGPPS*—geranylgeranyl diphosphate synthase; *SQS*—squalene synthase; *SMT-1*—sterol methyltransferase 1, *SDS/CYP710A*—sterol-22-desaturase; *CHS*—chalcone synthase, *F3′H*—flavonoid 3′-hydroxylase, *PAL*—phenylalanine ammonia-lyase, *UFGT*—UDP-glucose flavonoid 3-O-glucosyltransferase, *TAT*—tyrosine aminotransferase, *HPPR*—4 hydroxyphenylpyruvate reductase, *C4H*—cinnamic acid 4-hydroxylase, *4CL*—4-coumarate-CoA ligase, *RAS*—rosmarinic acid synthase.

**Table 3 ijms-25-12682-t003:** Comparison of plant (cell suspensions and HRs), microbial, and animal cells regarding their special characteristic correlated to bioreactors design and processes considerations according to [161,162,163,164].

Characteristics	Cell Suspension	Hairy Roots	Microbial Cells	Animal Cells
Shape	Spherical/cylindrical	Highly branched root tissues	Spherical/cylindrical	Spherical
Size, µM	10–200	10–200	1–10	10–100
Growth pattern	Small aggregates	Large aggregates	Individual cells/small aggregates	Support required for growth
Doubling time, h	20–100	20–100	0.5–4	24–48
Cultivation time	Days	Weeks	Weeks	Weeks
Cell aggregation	Cell clusters <100 μm > 2 mm	Cell clusters <100 μm > 2 mm	Single cells	Single cells
Shear sensitivity	High	High	Low	Very high
Inoculum size, %	5–10	5–10	1–2	5–10
Oxygen uptake rate, mmoL/L/h	2–10	2–10	10–200	0.05–10
Damage by aeration	Low	Low	Very low	High
Genetic and biochemical stability	Variable	Stable	Stable	Stable
Product localization	Intracellular	Intracellular	Extracellular/ Intracellular	Extracellular/ Intracellular
Posttranslational modifications	Yes	Yes	No	Yes
Required k_L_a value in bioreactor cultivation, h	10–150	10–50	100–1000	0.25–10
Culture scale-up	Easy	Difficult	Easy	Difficult

**Table 4 ijms-25-12682-t004:** Comparison of the major bioreactor types, used for HRs upscaling cultivation according to [28,159,167].

Bioreactor Type	Features and Modifications	Advantages	Disadvantages
Stirred tank (STR)	Equipped with impeller or turbine blades and sparger to facilitate mass transfer; modified impellers and the addition of mesh supports to avoid tissue damage are applied	Prevents cell aggregation; suitable for batch and continuous operation; improved nutrient and oxygen transfer compared to BCR	High shear forces and tissue damage; complicate configuration and increase exposure to contamination; high energy consumption and insufficient control of concentration gradients near dense root clumps
Bubble column (BCR)	Vertical cylindrical vessel with a sparger at the bottom for aeration and mixing; optionally, the vessel can be segmented and provided with mesh supports	Low shear stress; simple design, operation, and maintenance; low chance for contamination and energy requirements	Non-uniform fluid pattern; not suitable for high-viscosity liquids and high-density cell cultures; sedimentations of the cells; insufficient mass transfer due to gas flow channeling around the root clumps
Airlift (ALR)	Vertical cylindrical vessel with an external or internal draft tube (to improve the circulation and oxygen transfer) and a sparger at the bottom	Low shear stress; defined flow pattern due to the draft tube; enhanced mass transfer; less bubble coalescence	Greater air supply needed; formation of excessive foaming; higher energy requirements
Nutrient mist (NMR)	Consists of a growth chamber equipped with a mesh for HRs immobilization; a mist generator delivers medium in the form of mist generated by ultrasonic methods on the top (droplet sizes 0.01–10 μm), drained on the bottom of the vessel and recirculated again	Abundant oxygen and nutrient availability; better gas exchange and reduced shear stress	Complex construction; high energy consumption; labor-intensive setup
Trickle bed (TBR)	Has a similar structure to the NMR except the mist generator and the droplet size is bigger, >10 µm	Abundant oxygen supply; low energy consumption	May produce a viscous liquid film on the roots, creating a high mass transfer barrier; labor-intensive setup

**Table 5 ijms-25-12682-t005:** Examples of SMs biosynthesis and HRs cultivation in bioreactors.

Metabolite (s)	Plant Species	Bioreactor Volume and Configuration	Operating Conditions	Metabolite Content	Reference
Stirred tank reactor (STR)					
Boeravinone B Eupalitin	*Boerhaavia diffusa* L.	10-L; central-positioned nylon mesh	25 °C, 75 rpm agitation, 2.5 L/min flow rate	0.0299 mg/g DW 0.0031 mg/g DW	[171]
Saponins	*Panax ginseng* C.A.Mey.	7-L; cylindrical stainless-steel mesh	25 °C, 70 rpm agitation, 0.5 vvm flow rate	20.03 mg/g DW	[175]
Artemisinin	*Artemisia annua* L.	3-L; a perforated Teflon mesh at the bottom	25 °C, 125 rpm agitation, 0.25 vvm flow rate; 16/8 h light/dark; batch mode	4.63 mg/L	[173]
Artemisinin	*A. annua* L.	3-L; a perforated Teflon mesh at the bottom	25 °C, 125 rpm agitation, 0.25 vvm flow rate; 16/8 h light/dark; fed-batch mode	13.68 g/L	[174]
Curcumin	*Atropa belladonna* L.	10-L; central positioned mesh	25 °C, 75 rpm agitation, 2.5 L/min flow rate	0.112 mg/g DW	[176]
Azadirachtin	*Azadirachta indica* A. Juss.	3-L; polyurethane foam disk for support	26 °C, 0.2 vvm flow rate	97.28 mg/L	[177]
Indole alkaloids	*Vinca minor* L.	5-L; nylon mesh support	25 °C, 100 rpm agitation, 2.0 L/min flow rate, 3000 lx illumination	4.1 mg/g DW	[178]
Tropane alkaloids	*Brugmansia candida* Pers	1.5-L; a plastic mesh placed forming a zigzag arrangement around the baffle	24 °C, 50 rpm agitation, 0.5 vvm flow rate	3.72 mg/L	[179]
Artemisinin	*A. annua* L.	3-L; no modifications	25 °C, 125 rpm agitation, 40% dO_2_	0.32 mg/g	[180]
Picroliv	*Picrorhiza kurroa* Royle ex Benth.	5-L; no modifications	25 °C, 50 rpm agitation, 0.5 L/min flow rate	6.65 mg/g DW	[181]
Bubble column reactor (BCR)					
Saponins	*P. ginseng* C.A. Mey.	1.6-L; four compartment stages separated by stainless-steel mesh	25 °C, 0.5 vvm flow rate	19.89 mg/g DW	[175]
Artemisinin	*A. annua* L.	1.5-L; stainless-steel mesh	23 °C, 0.167 vvm flow rate	2.64 µg/g DW	[182]
Azadirachtin	*A. indica* L.	3-L; polyurethane foam disk for support	25 °C, 0.2 vvm flow rate	20.23 mg/L	[183]
Salicylic acid	*Hyoscyamus niger* L.	0.5-L; polypropylene mesh	25 °C, 670 mL/min flow rate; 12/12 h light/dark	60 µg/L	[184]
Scopolamine, Hyoscyamine Cuscohygrine	*H. niger* L.	0.6-L; immobilization basket in the centre	25 °C, 0.8 vvm flow rate	5.3 mg/g DW 1.6 mg/g DW 26.5 mg/g DW	[185]
Betalains	*Beta vulgaris* L.	3-L; plastic basket	23 °C, 33.4 cm^3^/s flow rate	280 mg/L	[186]
Betalains	*B. vulgaris* L.	3-L; plastic basket	23 °C, 33.4 cm^3^/s flow rate	36.13 mg/g DW	[187]
Ajmalicine	*C. roseus* (L.) G.Don	3-L; no modifications	23 °C, 0.3 vvm flow rate	15.5 mg/L	[188]
Ajmalicine	*C. roseus* (L.) G.Don	5-L; polypropylene mesh	23 °C, 0.3 vvm flow rate	30.0 mg/L	[188]
Ajmalicine	*C. roseus* (L.) G.Don	5-L; polyurethane foam	23 °C, 0.3 vvm flow rate	34.0 mg/L	[188]
Esculin	*Cichorium intybus* L.	1.75-L; no modifications	25 °C, 33.4 cm^3^/s flow rate	18.5 g/L	[189]
Azadirachtin	*A. indica* L.	3-L; no modifications	23 °C, 0.2 vvm flow rate	10.7 mg/L	[177]
Artemisinin	*A. annua* L.	3-L; perforated Teflon disc	25 °C, 0.3 vvm flow rate; 16/8 h light/dark	5.68 g/L	[190]
Airlift reactors (ALR)					
Monoterpenoid oxindole alkaloids	*Uncaria tomentosa* Willd	2-L; internal-loop draft tube	25 °C, 0.1 vvm flow rate; 15 µmol/m^2^/s continuous light	9.06 mg/L	[191]
Cichoric acid	*Echinacea purpurea* Moench	1.7-L; vertical stainless-steel mesh cylinder	25 °C, 0.002–0.004 m^3^/h flow rate; 600 µmol/m^2^/s continuous light	178.2 mg/L	[192]
Saponins	*Solanum chrysotrichum* Schltdl	2-L; internal loop draft-tube	26–27 °C, 0.1 vvm flow rate; 8–10 µmol/m^2^/s continuous light	0.7 mg/g DW	[193]
Astragaloside IV	*Astragalus mongholicus* Bunge	30-L; no modifications	26–27 °C, 400 mL/min flow rate	1.4 mg/g DW	[194]
Betacyanin	*B. vulgaris* L.	5-L cone-type ALR	25 °C, 0.35 vvm flow rate; 60 µmol/m^2^/s continuous light	34 mg/g DW	[195]
Nutrient mist reactor (NMR)					
Artemisinin	*A. annua* L.	5-L; stainless-steel mesh	23 °C, 0.167 vvm flow rate; two-stage operation: as a BCR and NMR; mist mode: 5/15 min on/off	2.64 µg/g DW	[182]
Esculin	*C. intybus* L.	1.75-L; two-tier nylon mesh	25 °C, 33.4 cm^3^/s flow rate	13.8 g/L	[189]
Azadirachtin	*A. indica* L.	3-L; no modifications	23 °C, 8.6 L/min flow rate	27.24 mg/L	[177]
Artemisinin	*A. annua* L.	3-L; perforated Teflon disc	25 °C, 0.3 vvm flow rate; 16/8 h light/dark; two-stage operation: as a BCR and NMR; mist mode: 5/30 s on/off	23.02 g/L	[190]
Rotating drum reactor (RDR)					
Ajmalicine	*C. roseus* (L.) G. Don	7-L; cylindrical mesh	23 °C, 0.3 vvm flow rate	4.6 mg/L	[188]
Hybrid bioreactors					
Anisodamine	*H. niger* L.	0.6-L; bubble-column/spray bioreactor	25 °C, 0.8 vvm flow rate (7 days working in a BCR mode and then switches to spraying mode)	0.67 mg/g DW	[185]

## Data Availability

Not applicable.

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
