# Peer review of "The Untapped Potential of Hairy Root Cultures and Their Multiple Applications"

_ijms, 2024, doi:10.3390/ijms252312682_

Round 1

Reviewer 1 Report

Comments and Suggestions for Authors

In this review, the authors have provided an in-depth review of the utility and potential of hairy root cultures in the production of specialized metabolites, and have successfully compiled a comprehensive analysis of the current state of hairy root cultures, their induction by Rhizobium rhizogenes, and the biotechnological tools available to enhance the biosynthesis of valuable secondary metabolites. The manuscript also delves into the design of bioreactors for culturing hairy roots and the process intensification technologies that can maximize their biosynthetic capacity. Furthermore, the review addresses the commercial and industrial implications, including examples of patented plant-derived products and the challenges and opportunities associated with scaling up the production process. The article is well-structured, informative, and contributes significantly to the field of plant biotechnology and metabolic engineering. However, there are several areas where further clarification and expansion could enhance the manuscript's impact and readability. Below are my detailed suggestions for improvement.

In the section 3.5.2. CRISPR/Cas9, I would like to see some discourse or examples of editing of non-coding regulatory sequences to achieve precise regulation of metabolites/products. The authors also mention the regulatory role of transcription factors, so gene editing of the motifs that transcription factors bind to could theoretically arrive at precise regulation, and these results are also an important aspect of the future, i.e., how to precisely regulate the amount of metabolites/end-products.

This related content also needs to be reflected in the Conclusions and Future Perspectives section.

Author Response

We would like to express our sincere thanks to yours’ and the reviewers’ constructive and helpful comments. The manuscript has been revised accordingly and our detailed responses to the comments are hence listed below. All changes made in the manuscript are highlighted with track changes. We hope that the revised version is now suitable for reproduction in International Journal of Molecular Science.

Reviewer #1

  • In this review, the authors have provided an in-depth review of the utility and potential of hairy root cultures in the production of specialized metabolites, and have successfully compiled a comprehensive analysis of the current state of hairy root cultures, their induction by Rhizobium rhizogenes, and the biotechnological tools available to enhance the biosynthesis of valuable secondary metabolites. The manuscript also delves into the design of bioreactors for culturing hairy roots and the process intensification technologies that can maximize their biosynthetic capacity. Furthermore, the review addresses the commercial and industrial implications, including examples of patented plant-derived products and the challenges and opportunities associated with scaling up the production process. The article is well-structured, informative, and contributes significantly to the field of plant biotechnology and metabolic engineering. However, there are several areas where further clarification and expansion could enhance the manuscript's impact and readability. Below are my detailed suggestions for improvement.

Author’s response: We deeply appreciate the expertise and time you contributed for reviewing our manuscript. We have carefully considered your suggestions and revised the text accordingly. Below, you will find our responses to your comments.

  • In the section 3.5.2. CRISPR/Cas9, I would like to see some discourse or examples of editing of non-coding regulatory sequences to achieve precise regulation of metabolites/products. The authors also mention the regulatory role of transcription factors, so gene editing of the motifs that transcription factors bind to could theoretically arrive at precise regulation, and these results are also an important aspect of the future, i.e., how to precisely regulate the amount of metabolites/end-products. This related content also needs to be reflected in the Conclusions and Future Perspectives section.

Author’s response: Thank you for this comment. We have now introduced more references and referred to several other works that have been conducted with regard to the regulatory potential of non-enzyme-coding DNA/RNA sequences. Examples of regulator TFs (positive and negative) are mentioned in the revised version and a collection of non-coding regulatory sequences with potential influence on secondary metabolites in medicinal plants have been referred to. In particular and in addition, we introduced the work of Rahul Mahadev Shelake et al., 2023 (https://doi.org/10.1016/j.plaphy.2023.108070) where they summarized 82 Non-coding RNAs (ncRNAs) associated with the regulation of plant secondary metabolite (SMs) production. This approach is now also added to the conclusion section as you suggested. The relevant correction have been marked with track changes option on Page 16, Lines: 501-522 and Page 34: Lines: 1096-1097 and Lines: 1102-1105.

We thank you again for your valuable comments and we do hope that the revised version is now meeting your expectation.

Reviewer 2 Report

Comments and Suggestions for Authors

The manuscript by Mirmazloum and colleagues deals with possible applications of a biotechnological system that has been investigated for several decades, namely hairy root cultures.

The main concern I have with this manuscript is its similarity to numerous other review papers, including recent ones, such as:

Morey, K. J., & Peebles, C. A. (2022). Hairy roots: An untapped potential for production of plant products. Frontiers in Plant Science13, 937095.

Another concern I have is the lack of information reported in section 2, where the natural process of hairy roots induction in planta and the biotechnological process for obtaining in vitro hairy root cultures are not well distinguished and are not described exhaustively. If the authors do not intend to consider these issues in depth, they can cite papers in which they are duly described.

In addition, the techniques for transforming plants based on A. rhizogenes are not presented. Again, if the authors do not intend to delve deeper into the topic, they should indicate the reference literature. It should also be reported that downstream of the transformation it is possible to obtain different biological systems, such as cell cultures, hairy root cultures and regenerated plants which can be propagated in vitro or transferred to the field. From Figure 1 it seems that the only aspect of hairy roots is the culture in a bioreactor.

In addition, the techniques for plant transformation based on A. rhizogenes are not presented. Again, if the authors do not intend to delve deeper into the topic, they should indicate the reference literature. It should also be reported that downstream of the transformation it is possible to obtain different biological systems, such as cell cultures, hairy root cultures and regenerated plants which can be propagated in vitro or transferred to the field. From Figure 1 it seems that the bioreactor culture is the only fate of hairy root cultures.

In the tables the genus of the scientific names of the species is often abbreviated. To avoid confusion, I recommend using the full name, at least the first time the genus is mentioned in the table. Also, as for the authority, I recommend always using the same format. For example, in Table 1 Papaver armeniacum (L.) appears and immediately after D. stramonium L. The authority is Linnaeus in both cases, but once the abbreviation is in brackets and the other time not. I recommend carefully checking the rules of species nomenclature in the entire manuscript.

In subsections 3.5 and 3.6 the experimental model used (plant, hairy root cultures, cell cultures, etc.) is often not mentioned. This is quite important since the results of metabolic engineering vary greatly depending on the system under experimentation. Furthermore, since this is a review on hairy roots, it would be preferable if the examples mainly refer to this system.

Unfortunately I cannot express an opinion on the section on bioreactors, as this topic is totally outside my area of ​​expertise.

Lastly, I suggest that the authors conduct a comprehensive revision of the manuscript with two primary objectives, namely to give the review originality by comparing it with others already published in recent literature, and to focus more on hairy root cultures.

Other minor doubts and recommended changes are indicated in the attached pdf file.

Author Response

We would like to express our sincere thanks to yours’ and the reviewers’ constructive and helpful comments. The manuscript has been revised accordingly and our detailed responses to the comments are hence listed below. All changes made in the manuscript are highlighted with track changes. We hope that the revised version is now suitable for reproduction in International Journal of Molecular Science.

Reviewer #2

The manuscript by Mirmazloum and colleagues deals with possible applications of a biotechnological system that has been investigated for several decades, namely hairy root cultures.

Author’s response: We express our sincere appreciation to you for investing your time and expertise to review our work. We have taken your suggestions carefully and revised the manuscript accordingly. Please find below our responses to your comments.

1) The main concern I have with this manuscript is its similarity to numerous other review papers, including recent ones, such as:

 Morey, K. J., & Peebles, C. A. (2022). Hairy roots: An untapped potential for production of plant products. Frontiers in Plant Science13, 937095.

Author’s response: Thank you for pointing this out. Our research shows that from 2022 up to now 74 review papers have been published according to Scopus with the keywords “hairy roots”. Most of these review papers have limited focus on hairy roots induction process, strategies to enhance their biosynthetic capacity and application. Other reviews focus mostly on the biotechnological process of hairy roots cultivation and their scale-up in general or are focused on a concrete plant species. The mentioned review of Morey, K. J., & Peebles, C. A. (2022). Hairy roots: An untapped potential for production of plant products. Frontiers in Plant Science, 13, 937095 is also cited in our manuscript. The latter review is briefly presented the hairy roots induction process and applications more focused on phytoremediation, production and isolation of secondary metabolites from hairy roots with examples mainly limited to phenolic compounds. Further, the paper is presenting in more details the production of terpenoid indole alkaloids from Catharanthus roseus hairy roots.

In our manuscript, we are aiming to present a comprehensive review on hairy roots and follow a thread starting from the hairy roots process induction, the available biotechnological tools to enhance the biosynthesis of valuable secondary metabolites supported with many examples. Along with that to better illustrate the potential of the molecular tools for engineering plant secondary metabolism, we have selected a case study that explores the potential strategies for manipulating the biosynthetic pathways of morphinan alkaloids at the level of structural genes and transcriptional factors. Further, we also go into depth of the bioreactor design for hairy root cultivation and the process intensification technologies to maximize their biosynthetic capacity. The challenges and opportunities associated with scaling up the production process are also presented. To give completeness of the review, we reveal the multiple applications of the hairy roots as a production system of many secondary metabolites at industrial scale and give examples of patented plant-derived products based on hairy roots cultivation.

A brief comparison of other reviews’ contents focused on HR cultures is also presented in page 2, lines: 83-88.

2) Another concern I have is the lack of information reported in section 2, where the natural process of hairy roots induction in planta and the biotechnological process for obtaining in vitro hairy root cultures are not well distinguished and are not described exhaustively. If the authors do not intend to consider these issues in depth, they can cite papers in which they are duly described.

Author’s response: Thank you for your valuable comment. We actually did not intend to go in depth of presenting the hairy roots induction in planta and in vitro, since we had different focus in this contribution. However, following your comment, we have inserted a short paragraph mentioning in brief the possibilities to induce hairy roots in planta and in vitro and gave links to reviews where natural process of hairy roots induction and the biotechnological process are described in more details. The text is added to page 3, lines 130-151.

 3) In addition, the techniques for transforming plants based on A. rhizogenes are not presented. Again, if the authors do not intend to delve deeper into the topic, they should indicate the reference literature. It should also be reported that downstream of the transformation it is possible to obtain different biological systems, such as cell cultures, hairy root cultures and regenerated plants which can be propagated in vitro or transferred to the field. From Figure 1 it seems that the only aspect of hairy roots is the culture in a bioreactor.

Author’s response: Thank you for your in-depth observation. We found this comment constructive and briefly mentioned the techniques for R. rhizogenes genetic transformation and listed the possibilities to obtain transgenic plants from hairy roots through spontaneous organ regeneration, somatic embryogenesis or organogenesis and field cultivation of transgenic plants. Although the idea behind the Figure 1 was to focus on the in vitro cultivation of the hairy roots in flasks and further scale-up in to bioreactors, we still found your comment very valuable and following your advice, the illustration of the Ri-plasmid is further modified for better visualization of its structure. We have now incorporated an enlarged representation of the Ri-plasmid in Figure 1. The modified version can be found in page 3, lines 130-151.

4) In addition, the techniques for plant transformation based on A. rhizogenes are not presented. Again, if the authors do not intend to delve deeper into the topic, they should indicate the reference literature. It should also be reported that downstream of the transformation it is possible to obtain different biological systems, such as cell cultures, hairy root cultures and regenerated plants which can be propagated in vitro or transferred to the field. From Figure 1 it seems that the bioreactor culture is the only fate of hairy root cultures.

Author’s response: Thank you for your comment. We believe that this paragraph has been duplicated by accident, since it seems to be the same as the above (number 3). Therefore we refer to our  answer as above.

5) In the tables the genus of the scientific names of the species is often abbreviated. To avoid confusion, I recommend using the full name, at least the first time the genus is mentioned in the table. Also, as for the authority, I recommend always using the same format. For example, in Table 1 Papaver armeniacum (L.) appears and immediately after D. stramonium L. The authority is Linnaeus in both cases, but once the abbreviation is in brackets and the other time not. I recommend carefully checking the rules of species nomenclature in the entire manuscript.

Author’s response: Thank you for correctly stating the inconsistencies. We agree with your comment and we have now modified the scientific names throughout the entire manuscript, including tables and unified the naming system.

6) In subsections 3.5 and 3.6 the experimental model used (plant, hairy root cultures, cell cultures, etc.) is often not mentioned. This is quite important since the results of metabolic engineering vary greatly depending on the system under experimentation. Furthermore, since this is a review on hairy roots, it would be preferable if the examples mainly refer to this system.

Author’s response: Thank you for pointing this out. We agree with your comment and relevant corrections in the mentioned sections have been done. The missing information of the experimental model used (plant, hairy root cultures, cell cultures, etc.) has been added. Indeed, all the experimental models used are hairy roots.

7) Unfortunately I cannot express an opinion on the section on bioreactors, as this topic is totally outside my area of ​​expertise.

Author’s response:  Thank you for your honest comment. We believe that the collected materials and presented examples have the potential to be considered by readers. A significant portion of this review is dedicated to highlight the conducted research and to present practical possibilities to initiate and manipulate the in vitro culture and secondary metabolites biosynthesis of hairy roots at large scale.

8) Lastly, I suggest that the authors conduct a comprehensive revision of the manuscript with two primary objectives, namely to give the review originality by comparing it with others already published in recent literature, and to focus more on hairy root cultures.

 Author’s response:  We have revised the manuscript accordingly. We made a short summary of the recently published reviews, which focus is different from ours or much more limited. We followed your recommendation and gave more focus on hairy roots by specifying the examples given in the manuscript. We agree with your comment that in section 3.5.3., there are examples that are not related to hairy roots and therefore have been removed. The changes took place in page 18-19, lines: 625-659.

9) Other minor doubts and recommended changes are indicated in the attached pdf file.

Author’s response: Thank you for your recommended changes and for your comprehensive reading of our manuscript, which we gladly accept. We believe that all your recommendations substantially increased the quality of the manuscript.

Page 1, line 19: “the” has been removed.

Page 1., line 34: It was explained that specialized metabolites can be named secondary metabolites as well.

Page 1-2, lines 43-46: The sentence has been reworded to “Some of the widely used plant-derived active ingredients with enormous commercial value that are incorporated in pharmacologically important formulations or solely administered are paclitaxel, atropine, morphine, dopamine and artemisinin”.

Page 2, line 57: Your comment is correct. The percentage significantly depends on the particular phytochemical. In our case the phytochemical is not defined, therefore not to make confuses “(less than 0.5%)” has been removed.

Page 2, line 61: The abbreviation “NPs” has been removed from the whole text. Instead, the full name “natural products” has been used all over the text.

Page 2, line 78: The word “heterogenity” was replaced with “somaclonal variation”.

Page 2, line 80: The repeating word “production” has been removed.

Page 3, line 107: The word “bacteria” has been changed to “bacterium”.

Page 2, line 108: We agree that the term “natural genetic engineer” refers to A. tumefaciens and therefore it has been removed.

Page 2, line 114: A diagram of Ri-plasmid was introduced in Figure 1.

Page 5, line 181: “Cell lines” has been replaced with “Root cultures with higher production capacity”.

Page 3, line 116: The term “plant genome” has been unified.

Page 3, line 119: The sentence has been corrected to “The T-DNA fragment contains a set of genes responsible for the synthesis of phytohormones, such as auxin and cytokinins which drives the characteristic development of HRs as well as genes encoding for opine synthesis (products of condensation of amino acids with ketoacids or sugars used by R. rhizogenes as a source of carbon and nitrogen) [16,17].”.

Page 6, lines 207-219 have been presented as a conclusion of section 2.

Page 6, lines 226-227: “cell lines” has been corrected to “selection of root cultures with higher production capacity”.

Page 6, line 236: The plant name has been corrected to N. tabacum L.

Page 7. Line 245: Reference 43 (Banerjee, S. Voyaging through Chromosomal Studies in Hairy Root Cultures towards Unravelling Their Relevance to Medicinal Plant Research: An Updated Review. The Nucleus 2018, 61 (1), 3–18. https://doi.org/10.1007/s13237-018-0227-x.) has been added.

Page 7, lines 260-262: The sentence has been corrected to “The maximum ginsenosides content (9 mg/g DW) in American ginseng (Panax quinquefolium C.A.Mey.) HRs was achieved in a liquid Gamborg B5 medium supplemented with 30 g/L sucrose [47].”.

Page 7, line 268: The plant species has been changed to Scutellaria baicalensis Georgi

Page 8, line 302: The plant species has been changed to Hyoscyamus reticulatus L

Page 8, lines 314-319: The sentence has been modified to “Elicitation is part of a plant’s defense mechanism. Elicitors are biotic (e.g. microorganisms, glycoproteins or polysaccharides derived from plant cell walls) or abiotic (e.g. chemical: organic or inorganic compounds of non-biological origin or physical: UV or low-energy ultrasound) factors that activate signal transduction pathways, leading to the induction or enhancement of SMs production in plant cells [60,61].”.

Page 8, lines 335: The plant name has been changed to “Datura stramonium L”.

Page 8, line 337: The additional space has been removed.

Page 9, line 346: The plant name has been changed to Astragalus membranaceus Bunge

Page 9, line 351: The additional interval has been removed.

Pages 9-11, Table 1: All plant species has been introduced with full names when mentioned for first time. The authority of each has been added.

Page 12, lines 361-362: The plant names has been changed to Calendula officinalis L. and Salvia miltiorrhiza Bunge

Pages 13-14, Table 2: All plant species has been introduced with full names when mentioned for first time. The authority of each has been added. The column “Strategy” has been united with the previous one under the name “Over expressed Genes/TFs”.

Page 14, line 314: The word “ecologanin” has been corrected to “secologanin”. The additional comma has been removed.

Page 15, line 469: The plant name has been changed to Phaseolus vulgaris L.

Page 15, line 473: The plant name has been changed to Mentha spicata L.

Page 16, line 489: The plant name has been changed to Atropa belladonna L.

Page 18-19, lines 625-659 has been removed, since they are not examples on hairy roots.

Round 2

Reviewer 2 Report

Comments and Suggestions for Authors

The authors have made the requested changes and have satisfactorily addressed this referee's concerns. In my opinion, the manuscript is now ready for publication.